SciPost Physics

Submission

# Feynman diagrams and $\Omega-$deformed M-theory

J. Oh[1*], Y. Zhou[2],

**1** Department of Physics, University of California, Berkeley, CA 94720, U.S.A.
**2** Perimeter Institute for Theoretical Physics, 31 Caroline St. N., Waterloo, ON N2L
2Y5, Canada
* jihwanoh@berkeley.edu

October 7, 2020

## Abstract

We derive the simplest commutation relations of operator algebras associated to M2 branes and an M5 brane in the $\Omega$-deformed M-theory, which is a natural set-up for Twisted holography. Feynman diagram 1-loop computations in the twisted-holographic dual side reproduce the same algebraic relations.

# 1    Introduction and Conclusions

In [1], Costello and Li developed a beautiful formalism, which prescribes a way to topo-
logically twist supergravity. Combining with the classical notion of topological twist of
supersymmetric quantum field theory [2,3], we are now able to explore a topological sector
for both sides of AdS/CFT correspondence. It was further suggested in [4] a systematic
method of turning an $\Omega$-background, which plays an important roles [5–10] in studying
supersymmetric field theories, in the twisted supergravity.

Topological twist along with $\Omega$-deformation enables us to study a particular protected
sub-sector of a given supersymmetric field theory [11–14], which is localized not only in
the field configuration space, but also in the spacetime. Interesting dynamics usually
disappear along the way, but as a payoff we can make more rigorous statement on the
operator algebra.

The topological holography is an exact isomorphism between the operator algebras of
gravity and field theory. In this paper, we will focus on a particular example of topological
holography: the correspondence of the operator algebra of M-theory on a certain back-
ground parametrized by $\epsilon_1$, $\epsilon_2$, which localizes to 5d non-commutative $U(K)$ Chern-Simons
theory with non-commutativity parameter $\epsilon_2$ [1], and the operator algebra of the worldvol-
ume theory of M2-brane, which localizes to 1d topological quantum mechanics(TQM). In
particular, [18] proved the isomorphism between two operator algebras. The isomorphism
was manifested by the mathematical notion, so called Koszul duality [18].

The important first step of the proof was to impose a BRST-invariance of the 5d $U(K)$
CS theory coupled with the 1d TQM. 5d CS theory is a renormalizable, and self-consistent
theory [17]. However, in the presence of the topological defect that couples 1d TQM and
5d CS theory, certain Feynman diagrams turn out to have non-zero BRST variations.
For the combined, interacting theory to be quantum mechanically consistent, the BRST
variations of the Feynman diagrams should combine to give zero. This procedure magically
reproduces the algebra commutation relations that define 1d TQM operator algebra, $\mathcal{A}_{\epsilon_1,\epsilon_2}$.
It is very intriguing that one can extract non-perturbative information in the protected
operator algebra from the perturbative calculation.

In fact, both the algebra of local operators in 5d CS theory and the 1d TQM opera-
tor algebra $\mathcal{A}_{\epsilon_1,\epsilon_2}$ are deformations of the universal enveloping algebra of the Lie algebra
$\mathrm{Diff}_{\epsilon_2}(\mathbb{C}) \otimes \mathfrak{gl}_K$ over the ring $\mathbb{C}[\![\epsilon_1]\!]$. Deformation theory tells us that the space of deforma-
tions of $U(\mathrm{Diff}_{\epsilon_2}(\mathbb{C}) \otimes \mathfrak{gl}_K)$ is the second Hochschild cohomology $\mathrm{HH}^2(U(\mathrm{Diff}_{\epsilon_2}(\mathbb{C}) \otimes \mathfrak{gl}_K))$.
Although this Hochschild cohomology is known to be hard to compute, there is still a clever

---

[1]The 5d CS theory that appear in this paper is always meant to be a certain variant of the usual 5d
CS theory with topological-holomorphic twist and with non-commutativity turned on in the holomorphic
directions.

way of comparing these two deformations [18]: notice that both of the algebras are defined compatibly for super groups $\mathrm{GL}_{K+R|R}$, and their deformations are compatible with transition maps $\mathrm{GL}_{K+R|R} \hookrightarrow \mathrm{GL}_{K+R+1|R+1}$, so they are actually controlled by elements in the limit

$$H^2(\lim_R \mathrm{HC}^*(U(\mathrm{Diff}_{\epsilon_2}(\mathbb{C}) \otimes \mathfrak{gl}_{K+R|R}))) \tag{1}$$

and the limit is well-understood, it turns out that the space of all deformations is essentially one-dimensional: a free module over $\mathbb{C}[\kappa]$ where $\kappa$ is the central element $1 \otimes \mathrm{Id}_K$. Hence the algebra of local operators in 5d CS theory and the 1d TQM operator algebra are isomorphic up to a $\kappa$-dependent reparametrization

$$\hbar \mapsto \sum_{i=1}^{\infty} f_i(\kappa) \hbar^i \tag{2}$$

where $f_i(\kappa)$ are polynomials in $\kappa$.

Later, in [19] the same algebra with $K = 1$ was defined using the gauge theory approach, and a combined system of M2-branes and M5-branes were studied. Especially, [19] interpreted the degrees of freedom living on M5-branes as forming a bi-module $\mathcal{M}_{\epsilon_1,\epsilon_2}$ of the M2-brane operator algebra, and suggested the evidence by going to the mirror Coulomb branch algebra [20, 21] and using the known Verma module structure of massive supersymmetric vacua [22, 23]. Appealing to the brane configuration in type IIB frame, they argued a triality in the M2-brane algebra, which can also be deduced from its embedding in the larger algebra, affine $gl(1)$ Yangian [24–27].

Crucially, [19] noticed $U(1)$ CS should be treated separately from $U(K)$ CS theory with $K > 1$, since the algebras differ drastically and the ingredients of Feynman diagram are different in $U(1)$ CS, due to the non-commutativity. As a result, the operator algebra isomorphism should be re-assessed.

Our work was motivated by the observation, and we will solve the following problems in a part of this paper.

- The simplest algebra $\mathcal{A}_{\epsilon_1,\epsilon_2}$ commutator, which has $\epsilon_1$ correction.

- Feynman diagrams whose non-trivial BRST variation lead to the simplest algebra commutator.

Next, we will make a first attempt to derive the bi-module structure from the 5d $U(1)$ CS theory, where the combined system of the M2-branes and the M5-brane is realized as the 1d TQM and the $\beta - \gamma$ system[2]. Especially, we will answer the following problems.

- The simplest algebra $\mathcal{A}_{\epsilon_1,\epsilon_2}$, bi-module $\mathcal{M}_{\epsilon_1,\epsilon_2}$ commutator, which has $\epsilon_1$ correction.

- Feynman diagrams whose non-trivial BRST variation lead to the simplest algebra $\mathcal{A}_{\epsilon_1,\epsilon_2}$, bi-module $\mathcal{M}_{\epsilon_1,\epsilon_2}$ commutator.

Our work is only a part of a bigger picture. The algebra $\mathcal{A}_{\epsilon_1,\epsilon_2}$ is a sub-algebra of affine $gl(1)$ Yangian [19], and there exists a closed form formula for the most general commutators, which can be derived from affine $gl(1)$ Yangian. One can try to derive the commutators from 5d $U(1)$ CS theory Feynman diagram computation.

Going to type IIB frame, the brane configurations map to Y-algebra configuration [28]. Here, the general M2-brane algebra is formed by the co-product of three different M2-brane algebras related by the triality. The local operators supported on M5-branes form a

---

[2]One way to understand the appearance of $\beta - \gamma$ system is to go to type IIA frame, where the M5-brane maps to a D4 brane, and the 11d supergravity background maps to a D6-brane. D4-D6 strings form 4d $\mathcal{N} = 2$ hypermultiplet. Under the $\Omega$-background, the 4d $\mathcal{N} = 2$ hypermultiplet localizes to $\beta\gamma$ system [11].

generalized $\mathcal{W}_{1+\infty}$ algebra [28]. The $\beta - \gamma$ Vertex Algebra that our M5-brane supports is the simplest example of this generalized $\mathcal{W}_{1+\infty}$. Hence, we are curious if our story can be further generalized to the coupled system of the 5d $U(1)$ CS theory and the generalized $\mathcal{W}_{1+\infty}$ algebra.

## 1.1 Structure of the paper

After reviewing the general concepts in section §2, we show the following algebra commutator in §3.1.

$$[t[2,1], t[1,2]]_{\epsilon_1} = \epsilon_1 \epsilon_2 t[0,0] + \epsilon_1 \epsilon_2^2 t[0,0] t[0,0] \tag{3}$$

where $[\bullet]_{\epsilon_1}$ is the $\mathcal{O}(\epsilon_1)$ part of $[\bullet]$, $t[m,n] \in \mathcal{A}_{\epsilon_1,\epsilon_2}$. The detail of the proof is shown in Appendix A.1. The commutation relation was successfully checked by 1-loop Feynman diagram associated to 5d CS theory and 1d TQM. This is the content of section §4. We collected some intermediate integral computations used in the Feynman diagram in Appendix B.1.

Next, we show the following algebra-bi-module commutator in §3.2.

$$\left[t[2,1], b[z^1]c[z^0]\right]_{\epsilon_1} = \epsilon_1 \epsilon_2 t[0,0] b[z^0] c[z^0] + \epsilon_1 \epsilon_2 b[z^0] c[z^0] \tag{4}$$

where $b[z^m]$, $c[z^m] \in \mathcal{M}_{\epsilon_1,\epsilon_2}$. The detail of the proof can be found in Appendix A.2. We reproduced the commutation relation using the 1-loop Feynman diagram computation in the 5d CS theory, 1d TQM, and 2d $\beta\gamma$ coupled system. This is the content of section §5. We collected some intermediate integral computations used in the Feynman diagram in Appendix B.2 and Appendix B.3.

*Note added: recently, complete commutation relations for the algebra $\mathcal{A}_{\epsilon_1,\epsilon_2}$ was proposed in [29].*

## 2 Twisted holography via Koszul duality

Twisted holography is the duality between the protected sub-sectors of full supersymmetric AdS/CFT [31–33], obtained by topological twist and $\Omega$-background both turned on in the field theory side and supergravity side. The most glaring aspect of twisted holography[3] is an exact isomorphism between operator algebra in both sides, which is manifested by a rigorous Koszul duality. Moreover, the information of physical observables such as Witten diagrams in the bulk side that match with correlation functions in the boundary side is fully captured by OPE algebra in the twisted sector [37].

This section is prepared for a quick review of twisted holography for non-experts. The idea was introduced in [1] and studied in various examples [4, 15, 18, 19, 38, 39] with or without $\Omega$-deformation. The reader who is familiar with [4] can skip most of this section, except for §2.2, §2.3, and §2.7, where we set up the necessary conventions for the rest of this paper. These subsections can be skipped as well, if the reader is familiar with [19]. Also, see a complementary review of the formalism in the section 2 of [19].

After defining the notion of twisted supergravity in §2.1, we will focus on a particular (twisted and $\Omega-$deformed) M-theory background on $\mathbb{R}_t \times \mathbb{C}_{NC}^2 \times \mathbb{C}_{\epsilon_1} \times \mathbb{C}_{\epsilon_2} \times \mathbb{C}_{\epsilon_3}$, where $NC$ means non-commutative, and $\epsilon_i$ stands for $\Omega-$background related to $U(1)$ isometry with a deformation parameter $\epsilon_i$ in §2.2. N $M2$ branes extending $\mathbb{R}_t \times \mathbb{C}_{\epsilon_1}$ leads to the field theory side. As we will explain in §2.3, a bare operator algebra isomorphism

---

[3]A similar line of development was made in [34,35], using twisted $\mathbb{Q}$-cohomology, where $\mathbb{Q}$ is a particular combination of a supercharge Q and a conformal supercharge S [36]. In the sense of [11], $\mathbb{Q}$-cohomology is equivalent to $Q_V$-cohomology, where $Q_V$ is the modified scalar super charge in $\Omega-$deformed theories.

between twisted supergravity and twisted M2-brane worldvolume theory is given by an interaction Lagrangian between two system. Due to this interaction, a perturbative gauge anomaly appears in various Feynman diagrams, and a careful cancellation of the anomaly will give a consistent quantum mechanical coupling between two systems. Strikingly, the anomaly cancellation condition itself leads to a complete operator algebra isomorphism, by fixing algebra commutators. This will be described in §2.5. To discuss holography, it is necessary to include the effect of taking large N limit and the subsequent deformation in the spacetime geometry. We will illustrate the concepts in §2.6. In §2.7, we will explain how to introduce M5-brane in the system and describe the role of M5-brane in the gravity and field theory side. In short, the degree of freedom on M5-brane will form a module of the operator algebra of M2-brane. Similar to M2-brane case, anomaly cancellation condition for M5-brane uniquely fixes the structure of the module.

## 2.1 Twisted supergravity

Before discussing the topological twist of supergravity, it would be instructive to recall the same idea in the context of supersymmetric field theory, and make an analogue from the field theory example.

Given a supersymmetric field theory, we can make it topological by redefining the generator of the rotation symmetry $M$ using the generator of the R-symmetry $R$.

$$M \quad \rightarrow \quad M' = M + R \tag{5}$$

As a part of Lorentz symmetry is redefined, supercharges, which were previously spinor(s), split into a scalar $Q$, which is nilpotent

$$Q^2 = 0, \tag{6}$$

and a 1-form $Q_\mu$. Because of the nilpotency of $Q$, one can define the notion of Q-cohomology.

Following anti-commutator explains the topological nature of the operators in Q-cohomology– a translation is Q-exact.

$$\{Q, Q_\mu\} = P_\mu \tag{7}$$

To go to the particular Q-cohomology, one needs to turn off all the infinitesimal super-translation $\epsilon_Q$ except for the one that parametrizes the particular transformation $\delta_Q$ generated by $Q$.

More precisely, if we were to start with a gauge theory, which is quantized with BRST formalism, the physical observables are defined as BRST cohomology, with respect to some $Q_{BRST}$. The topological twist modifies $Q_{BRST}$, and the physical observables in the resulting theory are given by $Q'_{BRST}$-cohomology.

$$Q_{BRST} \quad \rightarrow \quad Q'_{BRST} = Q_{BRST} + Q \tag{8}$$

As an example, consider $3d$ $\mathcal{N} = 4$ supersymmetric field theory. The Lorentz symmetry is $SU(2)_{Lor}$ and R-symmetry is $SU(2)_H \times SU(2)_C$, where H stands for Higgs and C stands for Coulomb. There are two ways to re-define the Lorentz symmetry algebra, and we choose to twist with $SU(2)_C$, as this will be used in the later discussion. In other words, one redefines

$$M \quad \rightarrow \quad M' = M + R_C \tag{9}$$

The resulting scalar supercharge is obtained by identifying two spinor indices, one of Lorentz symmetry $\alpha$ and one of $SU(2)_C$ R-symmetry $a$

$$Q_{a\dot{a}}^{\alpha} \quad \rightarrow \quad Q_{a\dot{a}}^{a} \tag{10}$$

and taking a linear combination.

$$Q = Q_{1\bar{1}}^{+} + Q_{1\bar{2}}^{-} \tag{11}$$

This twist is called Rozansky-Witten twist [40], and will be used in twisting our M2-brane theory.

One way to start thinking about the topological twist of supergravity is to consider a brane in the background of the "twisted" supergravity. If one places a brane in a twisted supergravity background, it is natural to guess that the worldvolume theory of the brane should also be topologically twisted coherently with the prescribed twisted supergravity background.

Given the intuition, let us define twisted supergravity, following [1]. In supergravity, the supersymmetry is a local(gauge) symmetry, a fermionic part of super-diffeomorphism. As usual in gauge theories, one needs to take a quotient by the gauge symmetry, and this is done by introducing a ghost field. As supersymmetry is a fermionic symmetry, the corresponding ghost field is a bosonic spinor, $q$. Twisted supergravity is defined as a supergravity in a background where the bosonic ghost $q$ takes a non-zero value.

It is helpful to recall how we twist a field theory to have a better picture for presumably unfamiliar non-zero bosonic ghost. One can think the infinitesimal super-translation parameter $\epsilon$ that appears in the global supersymmetry transformation as a rigid limit of the bosonic ghost $q$. For instance, in 4d $\mathcal{N} = 1$ holomorphically twisted field theory [41–44], with Q paired with $\epsilon_+$, the supersymmetry transformation of the bottom component $\phi$ of anti-chiral superfield $\bar{\Psi} = (\bar{\phi}, \bar{\psi}, \bar{F})$ transforms as

$$\delta\phi = \bar{\epsilon}\bar{\psi}, \quad \delta\bar{\psi} = i\epsilon_+\bar{\partial}\bar{\phi} + i\epsilon_-\partial\bar{\phi} + \bar{\epsilon}\bar{F} \tag{12}$$

As we focus on Q-cohomology, we set $\epsilon_+ = 1$, $\epsilon_- = \bar{\epsilon} = 0$, then the equations reduce into

$$\delta\bar{\phi} = 0, \quad \delta\bar{\psi} = i\bar{\partial}\bar{\phi} \tag{13}$$

In the similar spirit, in the twisted supergravity, we control the twist by giving non-zero VEV to components of the bosonic ghost $q$.

Indeed, [1] proved that by turning on non-zero bosonic spinor vacuum expectation value $\langle q \rangle \neq 0$ with $q_\alpha \Gamma_\mu^{\alpha\beta} q_\beta = 0$ for a vector gamma matrix, one can obtain the effect of topological twisting. We can now compare with the field theory case above (6): $Q^2 = 0$ with $Q \neq 0$. One can think of $\epsilon_Q$ as a rigid limit of q.

The operator algebra of twisted type IIB supergravity is isomorphic to that of Kodaira-Spencer theory [46]. The following diagram gives a pictorial definition of the two theories, which turned out to be isomorphic to each other.

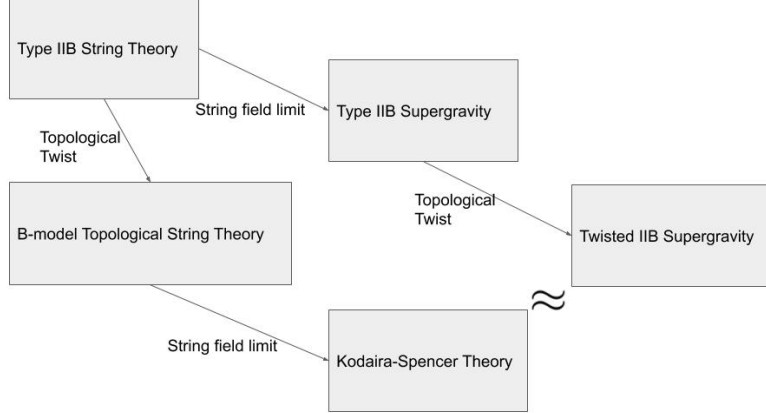

Figure 1: Starting from type IIB string theory, one can obtain same theory by taking two operations– 1. String field limit, 2. Topological twist– in any order.

Notice that the topological twist in the first column of the picture is the twist applied on the worldsheet string theory[4], whereas that in the second column is the twist on the target space theory.

Lastly, there are two types of twists available: a topological twist and a holomorphic twist, and it is possible to turn on the two different types of twists in the two different directions of the spacetime. The mixed type of twists is called a topological-holomorphic twist, e.g. [47]. Different from a topological twist, a holomorphic twist makes only the (anti)holomorphic translation to be Q-exact; after the twist we have $Q$ and $Q_z$ such that

$$\{Q, Q_z\} = P_{\bar{z}} \tag{14}$$

Hence, the holomorphic translation is actually physical(not Q-exact), and there exists non-trivial dynamics arising from this. [1,4] showed that it is possible to discuss a holomorphic twist in the supergravity. It is actually important to have a holomorphic direction to keep the non-trivial dynamics, as we will later see.

## 2.2 $\Omega$-deformed M-theory

Similar to the previous subsection, we will start reviewing the notion of $\Omega$-deformation of topologically twisted field theory. To define $\Omega$-background, one first needs an isometry, typically $U(1)$, generated by some vector field $V$ on a plane where one wants to turn on the $\Omega$-background. $\Omega$-deformation is a deformation of topologically twisted field theory and physical observables are defined with respect to the modified $Q_V$ cohomology, which satisfies

$$Q_V^2 = L_V, \quad \text{where } Q_V = Q + i_{V^\mu} Q_\mu \tag{15}$$

where $L_V$ is a conserved charge associated to $V$, and $i_{V^\mu}$ is a contraction with the vector field $V^\mu$, reducing the form degree by 1.

As the RHS of (15) is non-trivial, $Q_V$ cohomology only consists of operators, which are fixed by the action of $L_V-$ $\mathcal{O}$ such that $L_V \mathcal{O} = 0$. Hence, effectively, the theory is

---

[4]We thank Kevin Costello, who pointed out that the arrow from Type IIb string theory to B-model topological string theory is still mysterious in the following sense. In Ramond-Ramond formalism, as the super-ghost is in the Ramond sector and it is hard to give it a VEV. In the Green-Schwarz picture surely it should work better, but there are still problems there, as the world-sheet is necessarily embedded in space-time whereas in the B model that is not allowed.

defined in two less dimensions, if the isometry group is $U(1)$. More generally, one can turn on $\Omega$-background in the $n$ planes, and the dynamics of the original theory defined on $D$-dimensions is localized on $D - 2n$ dimensions.

In [4], a prescription for turning $\Omega$-background in twisted 11d supergravity was introduced; we need 3-form field $\epsilon C$, along with $U(1)$ isometry generated by a vector field $\epsilon V$, where $\epsilon$ is a constant, measuring the deformation. Similar to the field theory description, in this background($\langle q \rangle, C \neq 0$), the bosonic ghost $q$ squares into the vector field, $\epsilon V$ to satisfy the 11d supergravity equation of motion.

$$q^2 = q_\alpha (\Gamma^{\alpha\beta})_\mu q_\beta = \epsilon V_\mu \tag{16}$$

The $\Omega$-background localizes the supergravity field configuration into the fixed point of the $U(1)$ isometry. From now on, we will call $\Omega$-background with parametrized by $\epsilon_i$ as $\Omega_{\epsilon_i}$ background. More generally, one can turn on multiple $\Omega_{\epsilon_i}$-background in the separate 2-planes, which we will denote as $\mathbb{C}_{\epsilon_i}$.

The topologically twisted and $\Omega-$deformed 11d background that we will focus in this paper is

$$\text{11d SUGRA: } \mathbb{R}_t \times \mathbb{C}^2_{NC} \times \mathbb{C}_{\epsilon_1} \times TN_{1;\epsilon_2,\epsilon_3} \tag{17}$$

where $TN_{1;\epsilon_2,\epsilon_3}$ is Taub-NUT space, which can be thought of as $S^1_{\epsilon_2} \times \mathbb{R} \times \mathbb{C}_{\epsilon_3}$. The twist is implemented with the bosonic ghost chosen such that holomorphic twist in $\mathbb{C}^2_{NC}$ directions [5] and topological twist in $\mathbb{R}_t \times \mathbb{C}_{\epsilon_1} \times TN_{1;\epsilon_2,\epsilon_3}$ directions[6]. The 3-form is

$$C = V^d \wedge d\bar{z}_1 \wedge d\bar{z}_2 \tag{18}$$

where $V^d$ is 1-form, which is a Poincare dual of the vector field $V$ on $\mathbb{C}_{\epsilon_2}$ plane.

The statement of twisted holography is the duality between the protected subsector of M2-brane and the localized supergravity, due to the $\Omega$-background. We first want to introduce $M2$ branes and establish the explicit isomorphism at the level of operator algebras. Place $N$ M2-branes on

$$\text{M2-brane: } \mathbb{R}_t \times \{\cdot\} \times \mathbb{C}_{\epsilon_1} \times \{\cdot\} \tag{19}$$

To set up the stage for the concrete computation, it is convenient to go to type IIa frame by reducing along an M-theory circle. We pick the M-theory circle as $S^1_{\epsilon_2}$, which is in the direction of the vector field $V$.[7]

After reducing on $S^1_{\epsilon_2}$, the Taub-NUT geometry maps into one D6-brane and N M2-branes map to N D2-branes.

$$\begin{aligned}
\text{type IIa SUGRA : } & \mathbb{R}_t \times \mathbb{C}^2_{NC} \times \mathbb{C}_{\epsilon_1} \times \mathbb{R} \times \mathbb{C}_{\epsilon_3} \\
\text{D6-brane : } & \mathbb{R}_t \times \mathbb{C}^2_{NC} \times \mathbb{C}_{\epsilon_1} \\
\text{D2-branes : } & \mathbb{R}_t \times \qquad\quad \times \mathbb{C}_{\epsilon_1}
\end{aligned} \tag{20}$$

and 3-form C-field reduces into a B-field, which induces a non-commutativity $[z_1, z_2] = \epsilon_2$ on $\mathbb{C}^2_{NC}$.

$$B = \epsilon_2 d\bar{z}_1 \wedge d\bar{z}_2 \tag{21}$$

There are two types of contributions to gravity side: 1. closed strings in type IIa string theory and 2. open strings on the D6-brane. It was shown in [4] that we can completely

---

[5]NC stands for Non-Commutative. This will become clear in the type IIa frame.

[6]As remarked, if one introduces branes, the worldvolume theory inherits the particular twist that is turned on in the particular direction that the branes extend.

[7]For a different purpose, to make contact with Y-algebra system, type IIb frame is better, but we will not pursue this direction in this paper.

forget about the closed strings, so the open strings from the D6-brane entirely capture gravity side.

D6-brane worldvolume theory is 7d SYM, and it localizes on 5d non-commutative $U(1)$ Chern-Simons on $\mathbb{R}_t \times \mathbb{C}^2_{NC}$ due to $\Omega_{\epsilon_1}$-background on $\mathbb{C}_{\epsilon_1}$ [48]. The 5d Chern-Simons theory is not the typical Chern-Simons theory, as it inherits a topological twist in $\mathbb{R}_t$ direction and a holomorphic twist in $\mathbb{C}^2_{NC}$ direction, in addition to the non-commutativity. As a result, a gauge field only has 3 components

$$A = A_t dt + A_{\bar{z}_1} d\bar{z}_1 + A_{\bar{z}_2} d\bar{z}_2 \tag{22}$$

and the action takes the following form.

$$S = \frac{1}{\epsilon_1} \int_{\mathbb{R}_t \times \mathbb{C}^2_{NC}} dz_1 dz_2 \left( A \star dA + \frac{2}{3} A \star A \star A \right) \tag{23}$$

The star product $\star_{\epsilon_2}$ is the standard Moyal product induced from the non-commutativity of $\mathbb{C}^2_{NC}$: $[z_1, z_2] = \epsilon_2$. The Moyal product between two holomorphic functions[8] $f$ and $g$ is defined as

$$f \star_\epsilon g = fg + \epsilon \frac{1}{2} \epsilon_{ij} \frac{\partial}{\partial z_i} f \frac{\partial}{\partial z_j} g + \epsilon^2 \frac{1}{2^2 2!} \epsilon_{i_1 j_1} \epsilon_{i_2 j_2} \left( \frac{\partial}{\partial z_{i_1}} \frac{\partial}{\partial z_{i_2}} f \right) \left( \frac{\partial}{\partial z_{j_1}} \frac{\partial}{\partial z_{j_2}} g \right) \tag{24}$$

The gauge transformation $\Lambda \in \Omega^0(\mathbb{R} \times \mathbb{C}^2_{NC}) \otimes gl_1$[9] acting on the gauge field $A$ is

$$A \mapsto A + d\Lambda + [\Lambda, A], \text{ where } [\Lambda, A] = \Lambda \star_{\epsilon_2} A - A \star_{\epsilon_2} \Lambda \tag{25}$$

The field theory side is defined on $N$ D2-branes, which extend on $\mathbb{R}_t \times \mathbb{C}_{\epsilon_1}$. This is 3d $\mathcal{N} = 4$ gauge theory with 1 fundamental hypermultiplet and 1 adjoint hypermultiplet. Since the D2-branes are placed on topologically twisted supergravity background, the theory inherits the topological twist, which is Rozansky-Witten twist. We will work on $\mathcal{N} = 2$ notation, then each of $\mathcal{N} = 4$ hypermultiplet splits into a chiral and an anti-chiral $\mathcal{N} = 2$ multiplet. We denote the scalar bottom component of the fundamental chiral and anti-chiral multiplet as $I_a$ and $J^a$, and that of adjoint multiplets as $X^a_b$ and $Y^a_b$, where $a$ and $b$ are $U(N)$ gauge indices. Those scalars parametrize the hyper-Kahler target manifold $\mathcal{M}$, which has non-degenerate holomorphic symplectic structure. This structure turns the ring of holomorphic functions on $\mathcal{M}$ into a Poisson algebra with the following basic Poisson brackets:

$$\{I_a, J^b\} = \delta^b_a, \quad \{X^a_b, Y^c_d\} = \delta^a_d \delta^c_b \tag{26}$$

It is known that the gauge invariant combinations of Q-cohomology of Rozansky-Witten twisted $\mathcal{N} = 4$ theory is equivalent to the Higgs branch chiral ring. The elements of Higgs branch chiral ring are gauge invariant polynomials of $I$, $J$, $X$, and $Y$:

$$IS(X^m Y^n)J, \quad \mathrm{Tr}S(X^m Y^n) \tag{27}$$

where $S(\bullet)$ means fully symmetrized polynomial of the monomial $\bullet$.

Upon imposing the F-term relation[10]

$$[X, Y] + IJ = \epsilon_2 \delta, \tag{28}$$

---

[8]The Moyal product is extended to a product on the Dolbeault complex $\Omega^{0,*}(\mathbb{C}^2)$ by the same formula, except that the product between two functions becomes a wedge product between two forms.

[9]$gl_1$ Lie algebra factor comes from the simple fact that the theory is $U(1)$ gauge theory. For now, there is no essential difference between $\Omega^0(\mathbb{R} \times \mathbb{C})^2_{NC}$ and $\Omega^0(\mathbb{R} \times \mathbb{C})^2_{NC} \otimes gl_1$; however, having $gl_1$ rather than $gl_K$ makes a huge difference in the Feynman diagram computation, which will be discussed in §4.

[10]Physically, one needs to impose the F-term relation, as it is a part of defining condition for the supersymmetric vacua, as a critical locus of our specific 3d $\mathcal{N} = 4$ superpotential. Algebraically, F-term relation forms an ideal of the ring of holomorphic functions on $\mathcal{M}$.

one can show two words in (27) are equivalent[11] up to a factor of $\epsilon_2$[12], and the physical observables purely consist of one of them. Let us call them as

$$t[m,n] = \frac{1}{\epsilon_1}\text{Tr}SX^mY^n \tag{30}$$

In the $\Omega_{\epsilon_1}$-background, the Higgs branch chiral ring is quantized to an algebra and the support of the operator algebra in 3d $\mathcal{N} = 4$ theory also localizes to the fixed point of the $\Omega_{\epsilon_1}$-background. Therefore, the theory effectively becomes 1d TQM(Topological Quantum Mechanics) [23, 49, 50].

In summary, two sides of twisted holography are 5d non-commutative Chern-Simons theory and 1d TQM. Until now, we have not quite taken a large $N$ limit and resulting back-reaction that will deform the geometry. The large $N$ limit will be crucial for the operator algebra isomorphism to work and we will illustrate this point in the section §2.6.

## 2.3   Comparing elements of operator algebra

As 5d CS theory has a trivial equation of motion: $F = 0$, all the observables have positive ghost numbers. Also, since $\mathbb{R}_t$ direction is topological, the fields do not depend on $t$. As a result, operator algebra consist of ghosts $c(z_1, z_2)$ with holomorphic dependence on coordinates of $\mathbb{C}^2_{NC}$, $z_1$, $z_2$. The elements are then Fourier modes of the ghosts.

$$c[m,n] = \partial^m_{z_1}\partial^n_{z_2}c(0,0) \tag{31}$$

The non-commutativity in $\mathbb{C}^2_{NC}$ planes induces an algebraic structure in the holomorphic functions on $\mathbb{C}^2_{NC}$ defined by the Moyal product.

$$\left[z_1^a z_2^b, z_1^c z_2^d\right] = (z_1^a z_2^b) \star_{\epsilon_2} (z_1^c z_2^d) - (z_1^c z_2^d) \star_{\epsilon_2} (z_1^a z_2^b) = \sum_{m,n} f^{m,n}_{a,b;c,d} z_1^m z_2^n \tag{32}$$

At the classical level ($\epsilon_1 = 0$), the operator algebra $A_{\epsilon_1=0,\epsilon_2}$ of 5d CS theory is generated by ghost fields $c[m,n]$ with anti-commutativity relations, together with BRST differential $\delta$. As a graded associative algebra, $A_{0,\epsilon_2}$ is isomorphic to $\wedge^*\left(\mathbb{C}[z_1, z_2]_{\epsilon_2}\right) \cong \wedge^*\left(\text{Diff}_{\epsilon_2}\mathbb{C}\right)$, note that here we identify $z_1$ as $\partial_{z_2}$ using the Moyal product. The BRST differential $\delta$ is the dual of the Lie bracket, thus $A_{\epsilon_1=0,\epsilon_2}$ is the Chevalley-Eilenberg algebra of cochains on the Lie algebra $\mathfrak{g} = \text{Diff}_{\epsilon_2}\mathbb{C}\otimes\mathfrak{gl}_1$, denote by $C^*(\mathfrak{g})$. Note that here we treat the algebra $A_{\epsilon_1=0,\epsilon_2}$ as an algebra over the base ring $\mathbb{C}[\epsilon_1, \epsilon_2]$, so $\epsilon_1, \epsilon_2$ are algebraic parameters. At the quantum level, the operator algebra $A_{\epsilon_1=0,\epsilon_2}$ receives deformations, we will denoted it by $A_{\epsilon_1,\epsilon_2}$.

On the other hand, the elements of the algebra of operators in 1d TQM in the large $N$ limit consist of $t[m,n]$. The defining commutation relations come from the quantization of the Poisson brackets deformed by $\Omega_{\epsilon_1}$-background:

$$\left[I_a, J^b\right] = \epsilon_1\delta^b_a, \quad [X^a_b, Y^c_d] = \epsilon_1\delta^a_d\delta^c_b \tag{33}$$

We will write the F-term relation with explicit gauge indices as follows.

$$X^i_k Y^k_j - X^k_j Y^i_k + I_j J^i = \epsilon_2\delta^i_j \tag{34}$$

---

[11]They are related by following relation:

$$IS(X^mY^n)J = \epsilon_2\text{TrS}(X^mY^n) \tag{29}$$

[12]Note that the $\epsilon_2$ factor, which was previously introduced as a measure for the non-commutativity in the 5d CS theory, acts as an FI parameter in the 3d $\mathcal{N} = 4$ gauge theory.

We will call the algebra generated by $t[m,n]$ with relations (33), (34) as ADHM algebra or $\mathcal{A}_{\epsilon_1,\epsilon_2}$. Note that here we treat the algebra $\mathcal{A}_{\epsilon_1,\epsilon_2}$ as an algebra over the base ring $\mathbb{C}[\epsilon_1,\epsilon_2]$. This may seems to be strange at the first glance since $t[m,n]$ is *a priori* defined over $\mathbb{C}[\epsilon_1^{\pm},\epsilon_2]$, nevertheless the commutators between those $t[m,n]$'s only involve polynomials in $\epsilon_1$, so the algebra $\mathcal{A}_{\epsilon_1,\epsilon_2}$ is well-defined over $\mathbb{C}[\epsilon_1,\epsilon_2]$.

There is a one-to-one correspondence between $c[m,n]$ and $t[m,n]$, and [18] proved an isomorphism between ${}^!A_{\epsilon_1,\epsilon_2} = U_{\epsilon_1}(g)$ and $\mathcal{A}_{\epsilon_1,\epsilon_2}$ as $\mathbb{C}[\epsilon_1,\epsilon_2]$-algebras for 5d $U(K)$ Chern-Simons theory coupled with 1d TQM with $K > 1$, where ${}^!A_{\epsilon_1,\epsilon_2}$ is a Koszul dual of an algebra $A_{\epsilon_1,\epsilon_2}$[13]. The proof consists of two parts. First, one checks two algebras' commutation relations match in the $\mathcal{O}(\epsilon_1)$ order. Next, one proves the uniqueness of the deformation of the universal enveloping algebra $U(g)$ by $\epsilon_1$ that ensures all order matching. It worth mentioning that in the classical limit $\epsilon_1 \to 0$ the algebra $\mathcal{A}_{\epsilon_1,\epsilon_2}$ does not agree with the classical operator algebra of the ADHM mechanics, since the definition of the algebra $\mathcal{A}_{\epsilon_1,\epsilon_2}$ involves $1/\epsilon_1$, in other word, the isomorphism holds only at the quantum level.

One of our goal is to extend the $\mathcal{O}(\epsilon_1)$ order matching to $K = 1$. It may seem trivial compared to higher K, but it turns out that it is actually more complicated. We will give the proof in §4, §5. The uniqueness of the deformation applies for all $K$ including $K = 1$, so we will not try to spell out the details in this work.

## 2.4 Koszul duality

Let us explain why in the first place we can expect the Koszul duality between 5d and 1d operator algebra in the large $N$ limit. Further details on Koszul duality can be found in [19, 39, 51, 52]

The 5d theory is defined on $\mathbb{R}_t \times \mathbb{C}_{NC}^2$, where $\mathbb{R}_t$ is topological and $\mathbb{C}_{NC}^2$, and 1d TQM couples to the 5d theory along $\mathbb{R}_t$. As explained in (7), there is a scalar supercharge $Q$ and 1-form supercharge $\delta$ that anti-commute to give a translation operator $P_t$. We can build a topological line defect action using topological descent.

$$P\exp \int_{-\infty}^{\infty} [\delta, x(t)] \tag{35}$$

where

$$x(t) = \sum_{m,n} c[m,n]t[m,n] \tag{36}$$

The BRST variation of (35) vanishes if $x(t)$ satisfies a Maurer-Cartan equation:

$$[Q, x] + x^2 = 0 \tag{37}$$

and if $x \in A \otimes {}^!A$ for some $A$, the Maurer-Cartan equation is always satisfied. Hence, it is natural to expect the Koszul duality between $A_{\epsilon_1,\epsilon_2}$ and $\mathcal{A}_{\epsilon_1,\epsilon_2}$. So, the coupling between the 5d ghosts and gauge invariant polynomials of 1d TQM is given by

$$S_{int} = \int_{\mathbb{R}_t} t[m,n]c[m,n]dt. \tag{38}$$

Now that we have three types of Lagrangians:

$$S_{5d\ CS} + S_{1d\ TQM} + S_{int} \tag{39}$$

---

[13]It is known that for $A_{\epsilon_1=0,\epsilon_2} = C^*(g)$, the Koszul dual ${}^!A_{\epsilon_1=0,\epsilon_2}$ is $U(g)$ [45].

We need to make sure if the quantum gauge invariance of 5d Chern-Simons theory remains to be true in the presence of the interaction with 1d TQM. Namely, we need to investigate if there is non-vanishing gauge anomaly in Feynman diagrams. Along the way, we will derive the isomorphism between the operator algebras, as a consistency condition for the gauge anomaly cancellation.

## 2.5  Anomaly cancellation

To give an idea that the cancellation of the gauge anomaly of 5d CS Feynman diagrams fixes the algebra of operators in 1d TQM that couples to the 5d CS, let us review 5d $U(K)$ Chern-Simons example shown in [18]. Consider following Feynman diagram.

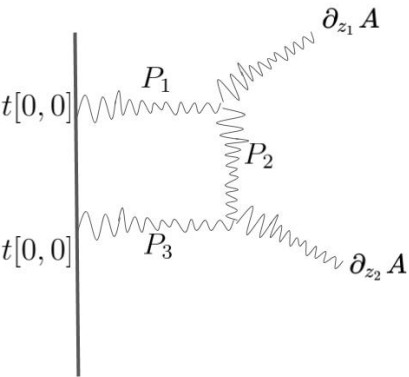

Figure 2: The vertical solid line represents the time axis. Internal wiggly lines stand for 5d gauge field propagators $P_i$, and the external wiggly lines stand for Fourier components 5d gauge field.

The BRST variation($\delta A = \partial c$) of the amplitude of the above Feynman diagram is non-zero.

$$\epsilon_1 \epsilon_{ij} (\partial_{z_i} A^a)(\partial_{z_j} c^b) K^{fe} f^c_{ae} f^d_{bf} t[0,0] t[0,0] \tag{40}$$

where $K^{ab}, f^a_{bc}$ are a Killing form and a structure constant of $u(K)$, and $t[m,n]$ is an element of ADHM algebra with gauge group $G = U(N)$, and flavor group $\hat{G} = U(K)$.

To have a gauge invariance, we need to cancel the anomaly, and the gauge variation of the following diagram has exactly factors like $\epsilon_{ij}(\partial_{z_i} A^a)(\partial_{z_j} c^b)$:

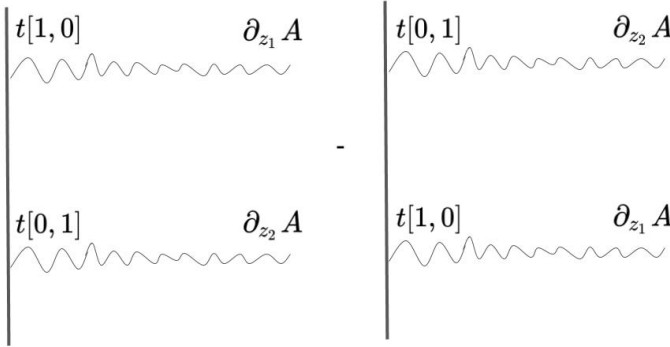

Figure 3:

The BRST variation of the amplitude is

$$\epsilon_1 \epsilon_{ij} (\partial_{z_i} A^a)(\partial_{z_j} c^b) K^{fe} f_{ae}^c f_{bf}^d [t[1,0], t[0,1]] \tag{41}$$

Imposing the cancellation of the BRST variation between (40) and (41), we obtain

$$[t[1,0], t[0,1]] = \epsilon_1 t[0,0] t[0,0] \tag{42}$$

This is very impressive, since we obtain the ADHM algebra from 5d Chern-Simons theory Feynman diagrams!

We will see that if $K = 1$, some ingredients of Feynman diagram change, but we can still reproduce ADHM algebra with $G = U(N)$, $\hat{G} = U(1)$.

## 2.6 Large $N$ limit and a back-reaction of $N$ M2-branes

Although we have not discussed explicitly about taking large $N$ limit, but it was being used implicitly in establishing the isomorphism between $^! A_{\epsilon_1,\epsilon_2}$ and $\mathcal{A}_{\epsilon_1,\epsilon_2}$.

Here we explain some detail of taking large $N$ limit. First notice that there are homomorphisms $\iota_N^{N'} : \mathcal{O}(T^*V_{K,N'}) \to \mathcal{O}(T^*V_{K,N})$ for all $N' > N$ induced by natural embedding $\mathbb{C}^N \hookrightarrow \mathbb{C}^{N'}$, where

$$V_{K,N} = \mathfrak{gl}_N \oplus \mathrm{Hom}(\mathbb{C}^K, \mathbb{C}^N), \tag{43}$$

so that $T^*V_{K,N}$ is the linear span of single operators $I, J, X, Y$, and the algebra $\mathcal{O}(T^*V_{K,N})$ is the commutative (classical) algebra generated by these operators (with no relations imposed). Then we define the *admissible* sequence of weight 0 as

$$\{f_N \in \mathcal{O}(T^*V_{K,N})^{\mathrm{GL}_N} | \iota_N^{N'}(f_{N'}) = f_N\}, \tag{44}$$

and for integer $r \geq 0$, a sequence $\{f_N\}$ is called admissible of weight $r$ if $\{N^{-r} f_N\}$ is admissible sequence of weight 0 (e.g. the sequence $\{N\}$ is admissible of weight 1), and define $\mathcal{O}(T^*V_{K,\bullet})^{\mathrm{GL}\bullet}$ to be the linear span of admissible sequences of all possible weights. It's easy to see that $\mathcal{O}(T^*V_{K,\bullet})^{\mathrm{GL}\bullet}$ is an algebra. Next we turn on the quantum deformation which turn the ordinary commutative product to the Moyal product $\star_{\epsilon_1}$, and it's easy to see that for admissible sequences $\{f_N\}$ and $\{g_N\}$, $\{f_N \star_{\epsilon_1} g_N\}$ is also admissible. In this way we obtained the quantized algebra $\mathcal{O}_{\epsilon_1}(T^*V_{K,\bullet})^{\mathrm{GL}\bullet}$.

The action of $\mathfrak{gl}_N$ on $V_{K,N}$ induces a moment map

$$\mu : \mathfrak{gl}_N \to \mathcal{O}_{\epsilon_1}(T^*V_{K,N}), \quad E_i^j \mapsto X_i^k Y_k^j - X_k^j Y_i^k + I_i J^j, \tag{45}$$

We want to set the moment map to $\epsilon_2$ times the identity, so we consider the shifted moment map:

$$\mu_{\epsilon_2} : \mathfrak{gl}_N \to \mathcal{O}_{\epsilon_1}(T^*V_{K,N}), \quad E_i^j \mapsto X_i^k Y_k^j - X_k^j Y_i^k + I_i J^j - \epsilon_2 \delta_i^j, \tag{46}$$

which is $\mathrm{GL}_N$-equivaraint. Together with the Moyal product on $\mathcal{O}_{\epsilon_1}(T^*V_{K,N})$, $\mu_{\epsilon_2}$ gives rise to a $\mathrm{GL}_N$-equivaraint map of left $\mathcal{O}_{\epsilon_1}(T^*V_{K,N})$-modules

$$\mu_{\epsilon_2} : \mathcal{O}_{\epsilon_1}(T^*V_{K,N}) \otimes \mathfrak{gl}_N \to \mathcal{O}_{\epsilon_1}(T^*V_{K,N}). \tag{47}$$

Taking $\mathrm{GL}_N$-invariance, we obtain the quantum moment map

$$\mu_{\epsilon_2} : (\mathcal{O}_{\epsilon_1}(T^*V_{K,N}) \otimes \mathfrak{gl}_N)^{\mathrm{GL}_N} \to \mathcal{O}_{\epsilon_1}(T^*V_{K,N})^{\mathrm{GL}_N}. \tag{48}$$

It's easy to varify that the image of $\mu_{\epsilon_2}$ is a two-sided ideal. Similar to $\mathcal{O}_{\epsilon_1}(T^*V_{K,\bullet})^{\mathrm{GL}\bullet}$, we can define admissible sequences in $(\mathcal{O}_{\epsilon_1}(T^*V_{K,N}) \otimes \mathfrak{gl}_N)^{\mathrm{GL}_N}$ and call this space $(\mathcal{O}_{\epsilon_1}(T^*V_{K,\bullet}) \otimes \mathfrak{gl}_\bullet)^{\mathrm{GL}\bullet}$. Quantum moment maps for all $N$ give rise to

$$\mu_{\epsilon_2} : (\mathcal{O}_{\epsilon_1}(T^*V_{K,\bullet}) \otimes \mathfrak{gl}_\bullet)^{\mathrm{GL}\bullet} \to \mathcal{O}_{\epsilon_1}(T^*V_{K,\bullet})^{\mathrm{GL}\bullet}, \tag{49}$$

and the image is a two-sided ideal, so we can take the quotient of $\mathcal{O}_{\epsilon_1}(T^*V_{K,\bullet})^{\mathrm{GL}\bullet}$ by this ideal, this is by definition the large-$N$ limit denoted by $\mathcal{O}_{\epsilon_1}(\mathcal{M}_{K,\bullet}^{\epsilon_2})$.

From the definition above, we can write down a set of generators of $\mathcal{O}_{\epsilon_1}(\mathcal{M}_{K,\bullet}^{\epsilon_2})$:

$$\{N\} \text{ and } \{I_\alpha S(X^n Y^m) J^\beta\} \text{ for all integers } n, m \geq 0. \tag{50}$$

Note that Costello also defined a combinatorical algebra $\mathcal{A}_{\epsilon_1,\epsilon_2}^{\mathrm{comb}}$ in section 10 of [18], which depends on $K$ but not on $N$. This is related to $\mathcal{O}_{\epsilon_1}(\mathcal{M}_{K,\bullet}^{\epsilon_2})$ in the sense that generators of $\mathcal{A}_{\epsilon_1,\epsilon_2}^{\mathrm{comb}}$ are

$$\{N\} \text{ and } \{\frac{1}{\epsilon_1} I_\alpha S(X^n Y^m) J^\beta\} \text{ for all integers } n, m \geq 0, \tag{51}$$

when $\epsilon_1 \neq 0$. In the notation of [18] they corresponds to

$$D(\emptyset) \text{ and } \mathrm{Sym}(D(\alpha \Downarrow, \uparrow^n, \downarrow_m, \beta \Uparrow)) \text{ for all integers } n, m \geq 0, \tag{52}$$

respectively. Under the aforementioned correspondence between generators, $\mathcal{A}_{\epsilon_1,\epsilon_2}^{\mathrm{comb}}$ is isomorphic to $\mathcal{O}_{\epsilon_1}(\mathcal{M}_{K,\bullet}^{\epsilon_2})$ (Proposition 13.4.3 of [18]) when $\epsilon_1$ is invertible.

The general philosophy of AdS/CFT [31] teaches us that the back-reaction of $N$ M2-branes will deform the spacetime geometry. In our case, since the closed strings completely decouple from the analysis, the back-reaction is only encoded in the interaction related to the open strings. More precisely, the back-reaction is already encoded in the 5d-1d interaction Lagrangian (38), a part of which we reproduce below.

$$S_{back} = \int_{\mathbb{R}_t} t[0,0]c[0,0]dt. \tag{53}$$

Here, we can explicitly see $N$ in $t[0,0]$, as

$$t[0,0] = IJ/\epsilon_1 = \epsilon_2 Tr\delta_j^i/\epsilon_1 = N\frac{\epsilon_2}{\epsilon_1} \tag{54}$$

where in the second equality, we used the F-term relation.

After taking large $N$ limit, $N$ becomes an element of the algebra $\mathcal{A}_{\epsilon_1,\epsilon_2}$, which is coupled to the zeroth Fourier mode of the 5d ghost, $c[0,0]$.

## 2.7 M5-brane in $\Omega-$deformed M-theory

We want to include one M5(D4)-brane in the story, and review the role played by the new element coming from the bi-module on M5(D4)-brane in the boundary and the bulk.

| | 0 | 1 | 2 | 3 | 4 | 5 | 6 | 7 | 8 | 9 | 10 |
|---|---|---|---|---|---|---|---|---|---|---|---|
| Geometry | $\mathbb{R}_t$ | $\mathbb{C}_{\epsilon_1}$ | | $\mathbb{C}_{NC}^2$ | | | | $\mathbb{C}_{\epsilon_3}$ | | $\mathbb{R}$ | $S_{\epsilon_2}^1$ |
| $M2(D2)$ | $\times$ | $\times$ | $\times$ | | | | | | | | |
| $M5$ | | $\times$ | $\times$ | $\times$ | $\times$ | | | | | $\times$ | $\times$ |
| $D4$ | | $\times$ | $\times$ | $\times$ | $\times$ | | | | $\times$ | | |

Table 1: M2, M5-brane

In the boundary perspective, it intersects with the M2(D2)-brane with two directions and supports 2d $\mathcal{N} = (2,2)$ supersymmetric field theory with two chiral superfields, whose bottom components are $\varphi, \tilde{\varphi}$, arising from $D2 - D4$ strings. This 2d theory interacts with the 3d $\mathcal{N} = 4$ ADHM theory with a superpotential

$$\mathcal{W} = \tilde{\varphi} X \varphi \tag{55}$$

where $X$ is a scalar component of the adjoint hypermultiplet of the 3d theory.

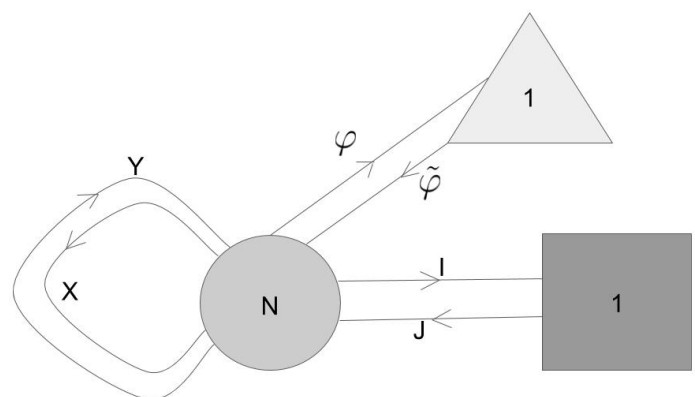

Figure 4: 3d $\mathcal{N} = 4$ ADHM quiver gauge theory with $G = U(N)$, $F = U(1)$, decorated with 2d $\mathcal{N} = (2,2)$ field theory. $X$, $Y$ are scalars of adjoint hypermultipet, and $I$, $J$ are scalars of (anti)fundamental hypermultiplet. The triangle node encodes the 2d theory. $\varphi$ and $\tilde{\varphi}$ are 2d scalars. In type IIA language, the circle, square, and triangle node correspond to D2, D6, D4 branes, respectively.

A naive set of gauge invariant operators living on the 2d intersection are

$$IX^m Y^n \tilde{\varphi}, \quad \varphi X^m Y^n J, \quad \varphi X^m Y^n \tilde{\varphi} \tag{56}$$

The superpotential reduces [19, 22] the above set into

$$\mathcal{M}_{\epsilon_1, \epsilon_2} = \{b[z^n] = IY^n \tilde{\varphi}, \quad c[z^n] = \varphi Y^n J\} \tag{57}$$

The set of 2d observables $\mathcal{M}_{\epsilon_1, \epsilon_2}$ forms a bi-module of the ADHM algebra $\mathcal{A}_{\epsilon_1, \epsilon_2}$.

The difference between left and right actions of the algebra $\mathcal{A}$ on $\mathcal{M}_{\epsilon_1, \epsilon_2}$ is encoded in the form of a commutator:

$$[a, m] = m', \quad \text{where } a \in \mathcal{A}, \quad m, m' \in \mathcal{M}_{\epsilon_1, \epsilon_2} \tag{58}$$

To verify (58), we need to establish the commutation relations between the set of letters $\{\varphi, \tilde{\varphi}\}$ and $\{X, Y, I, J\}$. Those are given by

$$\begin{aligned}
IP(\varphi, \tilde{\varphi}) &= P(\varphi, \tilde{\varphi})I \\
JP(\varphi, \tilde{\varphi}) &= P(\varphi, \tilde{\varphi})J \\
X_j^i P(\varphi, \tilde{\varphi}) &= P(\varphi, \tilde{\varphi})X_j^i \\
Y_j^i P(\varphi, \tilde{\varphi}) &= P(\varphi, \tilde{\varphi})(Y_j^i + \tilde{\varphi}^i \varphi_j) \\
X_j^i \varphi_i P(\varphi, \tilde{\varphi}) &= -\epsilon_1 \partial_{\tilde{\varphi}_j} P(\varphi, \tilde{\varphi}) \\
X_j^i \tilde{\varphi}^j P(\varphi, \tilde{\varphi}) &= -\epsilon_1 \partial_{\varphi_i} P(\varphi, \tilde{\varphi})
\end{aligned} \tag{59}$$

Again, the non-trivial commutation relations in the last three lines originates from the effect of the particular superpotential $\mathcal{W}$. For the derivation, we refer the reader to [19, 22].

In the $\Omega_{\epsilon_1}$-background, 2d $\mathcal{N} = (2,2)$ theory localizes to a point, which is the origin of $\mathbb{R}_t$.

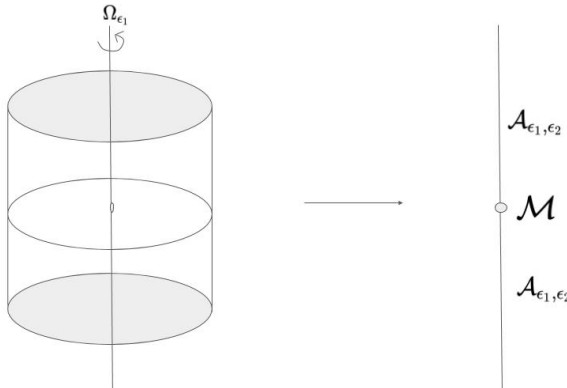

Figure 5: Left figure represents a coupled system of 3d $\mathcal{N} = 4$ ADHM theory(the cylinder) and 2d $\mathcal{N} = (2,2)$ theory(the middle disk in the cylinder) from D2 branes and a D4 brane. In the $\Omega_{\epsilon_1}$-background, the system localizes to $1d + 0d$ system.

Hence, the resulting system is ADHM algebra $\mathcal{A}_{\epsilon_1,\epsilon_2}$ and bi-module $\mathcal{M}_{\epsilon_1,\epsilon_2}$ of the algebra.

To study the bulk perspective, we need to study what degree of freedoms that M5-brane support in the 5d spacetime $\mathbb{R}_t \times \mathbb{C}^2_{NC}$ and how the M5-brane interacts with 5d Chern-Simons theory. 5d CS theory is defined in the context of type IIa, and M5-brane is mapped to a D4-brane. The local degree of freedom comes from D4-D6 strings, which are placed on $\{\cdot\} \times \mathbb{C} \in \mathbb{R}_t \times \mathbb{C}^2_{NC}$. These 2d degrees of freedom are actually coming from 4d $\mathcal{N} = 2$ hypermultiplet, as the true intersection between D4 and D6 is $\mathbb{C} \times \mathbb{C}_{\epsilon_1}$. In the $\Omega_{\epsilon_1}$-background, the 4d $\mathcal{N} = 2$ hypermultiplet localizes to a $\beta - \gamma$ system [11]. Hence, we arrive at $\beta - \gamma$ Vertex Algebra on $\mathbb{C} \subset \mathbb{C}^2_{NC}$.

|          | 0        | 1        | 2 | 3       | 4 | 5 | 6 | 7        | 8 | 9             |
|----------|----------|----------|---|---------|---|---|---|----------|---|---------------|
| Geometry | $\mathbb{R}_t$ | $\mathbb{C}_{\epsilon_1}$ |   | $\mathbb{C}^2_{NC}$ |   |   |   | $\mathbb{C}_{\epsilon_3}$ |   | $\mathbb{R}_{\epsilon_2}$ |
| 1d TQM   | $\times$ |          |   |         |   |   |   |          |   |               |
| 2d $\beta\gamma$ |    |          |   | $\times$ | $\times$ |   |   |          |   |               |
| 5d CS    | $\times$ |          |   | $\times$ | $\times$ | $\times$ | $\times$ |          |   |               |

Table 2: Bulk perspective

The $\beta - \gamma$ system minimally couples to 5d Chern-Simons theory via

$$\int_{\mathbb{C}} \beta(\bar{\partial} + A\star)\gamma \tag{60}$$

The observables to be compared with those of field theory side: $b[z^n]$ and $c[z^n]$ can be naturally compared with the modes of $\beta$ and $\gamma$: $\partial_z^n \beta$, $\partial_z^n \gamma$, and the Koszul duality manifests itself by the coupling between two types of observables:

$$\int_{\{0\}} \partial_{z_2}^{k_1} \beta \cdot b[z^{k_1}] + \int_{\{0\}} \partial_{z_2}^{k_2} \gamma \cdot c[z^{k_2}] \tag{61}$$

where $z = z_2$, and the integral on a point is merely for a formal presentation.

The following figure depicts the entire bulk and boundary system including the line and the surface defect, and describes how all the ingredients are coupled.

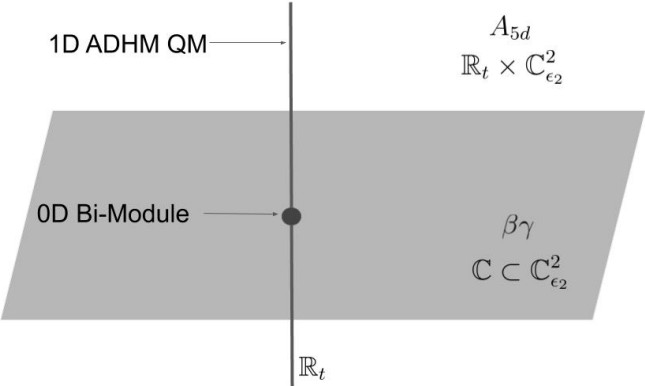

Figure 6: 5d Chern-Simons($\mathbb{R}_t \times \mathbb{C}^2_{NC}$), 1d generalized Wilson line defect($\mathbb{R}_t$), and 2d surface defect($\mathbb{C} \subset \mathbb{C}^2_{NC}$).

As explained in section §2.5, we need to make sure if the introduction of the 2d system is quantum mechanically consistent, or anomaly free. Imposing the anomaly cancellation condition of 5d, 2d, 1d coupled system, we should be able to derive the bi-module commutation relations defined in the field theory side. This is the content of §5.

# 3    M2-brane algebra and M5-brane module

In this section, we will provide a representative commutation relation for the algebra $\mathcal{A}_{\epsilon_1,\epsilon_2}$

$$\left[a, a'\right] = a_0 + \epsilon_1 a_1 + \epsilon_1^2 a_2 + \dots, \quad \text{where } a, a', a_i \in \mathcal{A}_{\epsilon_1,\epsilon_2} \tag{62}$$

and a representative commutation relation for the algebra $\mathcal{A}_{\epsilon_1,\epsilon_2}$ and the bi-module $\mathcal{M}_{\epsilon_1,\epsilon_2}$ for $\mathcal{A}_{\epsilon_1,\epsilon_2}$.

$$[a, m] = m_0 + \epsilon_1 m_1 + \epsilon_1^2 m_2 + \dots, \quad \text{where } a \in \mathcal{A}_{\epsilon_1,\epsilon_2}, \ m, m_i \in \mathcal{M}_{\epsilon_1,\epsilon_2} \tag{63}$$

We first recall the notation for a typical element of $\mathcal{A}_{\epsilon_1,\epsilon_2}$ and $\mathcal{M}_{\epsilon_1,\epsilon_2}$:

$$t[m, n] = \frac{1}{\epsilon_1} TrS(X^m Y^n) = \frac{1}{\epsilon_1 \epsilon_2} IS(X^m Y^n)J \in \mathcal{A}_{\epsilon_1,\epsilon_2}$$
$$b[z^m] = \frac{1}{\epsilon_1} IY^m \tilde{\varphi} \in \mathcal{M}_{\epsilon_1,\epsilon_2} \tag{64}$$
$$c[z^n] = \frac{1}{\epsilon_1} \varphi Y^n J \in \mathcal{M}_{\epsilon_1,\epsilon_2}$$

For the convenience of later discussions, we also introduce the notation:

$$T[m, n] = \frac{\epsilon_2}{\epsilon_1} TrS(X^m Y^n) = \frac{1}{\epsilon_1} IS(X^m Y^n)J \in \mathcal{A}_{\epsilon_1,\epsilon_2} \tag{65}$$

Our final goal is to reproduce the $\mathcal{A}_{\epsilon_1,\epsilon_2}$ algebra from the anomaly cancellation of 1-loop Feynman diagrams in 5d Chern-Simons theory. So, it is important to have commutation relations that yield $\mathcal{O}(\epsilon_1)$ term in the right hand side, where $\epsilon_1$ is a loop counting parameter in 5d CS theory.

## 3.1    M2-brane algebra

Since we have not provided a concrete calculation until now, let us give a simple computation to give an idea of ADHM algebra and its bi-module. It is useful to recall $G = U(N)$,

$\hat{G} = U(K)$ ADHM algebra, which serves as a practice example, and at the same time as an example that explains the non-triviality of $G = U(N)$, $\hat{G} = U(1)$ ADHM algebra, compared to $K > 1$ cases.

It was shown in [18] that following commutation holds for $G = U(N)$, $\hat{G} = U(K)$ ADHM algebra.

$$[t[1,0], t[0,1]] = \epsilon_1 t[0,0]t[0,0] \quad \text{or} \quad [IXJ, IYJ] = \epsilon_1(IJ)(IJ) \tag{66}$$

This does not work for $\hat{G} = U(1)$. It is instructive to see why.

$$\begin{aligned}[TrX, TrY] = [X_i^i, Y_j^j] &= \delta_j^i \delta_i^j \epsilon_1 = \delta_j^i \epsilon_1 \\ &= N\epsilon_1 \end{aligned} \tag{67}$$

Multiplying both sides by $\epsilon_2^2/\epsilon_1^2$, we can convert it into $T[m,n]$ basis:

$$[T[1,0], T[0,1]] = \epsilon_2 T[0,0] \tag{68}$$

The RHS of (68) is different from (66) crucially in its dependence on $\epsilon_1$. The RHS of (68) is $\mathcal{O}(\epsilon_1^0)$, but that of (66) is $\mathcal{O}(\epsilon_1)$. While it was sufficient to consider this simple commutator to see the $\epsilon_1$ deformation of the algebra for $\hat{G} = U(K)$ with $K > 1$, we need to consider a more complicated commutator to see $\mathcal{O}(\epsilon_1)$ correction in the RHS.

In Appendix §A.1, we will derive a set of relations that will determine all other relations, of which the simplest ones are:

$$\begin{aligned} [t[3,0], t[0,3]] &= 9t[2,2] + \frac{3}{2}\big(\sigma_2 t[0,0] - \sigma_3 t[0,0]t[0,0]\big) \\ [t[2,1], t[1,2]] &= 3t[2,2] - \frac{1}{2}\big(\sigma_2 t[0,0] - \sigma_3 t[0,0]t[0,0]\big) \end{aligned} \tag{69}$$

where

$$\sigma_2 = \epsilon_1^2 + \epsilon_2^2 + \epsilon_1\epsilon_2, \quad \sigma_3 = -\epsilon_1\epsilon_2(\epsilon_1 + \epsilon_2) \tag{70}$$

To compare the commutation relation to that from 5d Chern-Simons calculation, we need to make sure if the parameters of ADHM algebra $\mathcal{A}_{\epsilon_1, \epsilon_2}$ are the same as those in 5d CS theory. From [18], the correct parameter dictionary[14] is

$$(\epsilon_1)_{ADHM} = (\epsilon_1)_{CS}, \quad \left(\epsilon_2 + \frac{1}{2}\epsilon_1\right)_{ADHM} = (\epsilon_2)_{CS} \tag{71}$$

Hence, the commutation relation that we are supposed to match from the 5d computation is

$$[t[2,1], t[1,2]] = 3t[2,2] - \frac{1}{2}\left(\left(\epsilon_2^2 + \frac{3}{4}\epsilon_1^2\right)t[0,0] + \left(\epsilon_1\epsilon_2^2 - \frac{\epsilon_1^3}{4}\right)t[0,0]t[0,0]\right) \tag{72}$$

There is one term in the RHS of (72) that is in $\mathcal{O}(\epsilon_1)$ order:

$$[t[2,1], t[1,2]] = \mathcal{O}(\epsilon_1^0) - \frac{1}{2}\epsilon_1\epsilon_2^2 t[0,0]t[0,0] + \mathcal{O}(\epsilon_1^2) \tag{73}$$

We will try to recover the $\mathcal{O}(\epsilon_1)$ term from 5d Feynman diagram calculation[15] in section §4; the general argument that gauge anomaly cancelation leads to the Koszul dual algebra commutation relation is given in §2.5.

---

[14] We thank Davide Gaiotto, who pointed out this subtlety.

[15] The basis used in the Feyman diagram computation is $T[m,n]$, not $t[m,n]$. However, the change of basis does not affect any computation because the $\mathcal{O}(\epsilon_1)$ term in (73) is quadratic in $t$.

## 3.2   M5-brane module

We will use the commutation relations (33), (34), (59) to compute the commutators be-tween $a \in \mathcal{A}_{\epsilon_1,\epsilon_2}$ and $m \in \mathcal{M}_{\epsilon_1,\epsilon_2}$, which are defined in (30), (57). When one tries to compute some commutators, one immediately notices some normal ordering ambiguity in a general module element $m$, which can be seen in following example.

$$[IXJ, (I\tilde{\varphi})(\varphi J)] = \left[ I_i X^i_j J^j, I_a \tilde{\varphi}^a \varphi_b J^b \right] \tag{74}$$

Assuming that the order of letters is consistent with the order of fields in the real line $\mathbb{R}_t$, it is obvious that we need to place $\tilde{\varphi}^a \varphi_b$ together, as they are defined at a point $\{0\} \in \mathbb{R}_t{}^{16}$. However, it is ambiguous whether we put $I_a$, $J^b$ in the right or left of $\tilde{\varphi}^a \varphi_b$, as $I_a$, $J^b$ are living on $\mathbb{R}_t$. We will try to fix this ambiguity to prepare a concrete calculation.

Considering following normal ordering when writing a module element $(IY\varphi)(\varphi J)$ will be enough to fix the ambiguity.

$$|\tilde{\varphi}^j \varphi_k| I_i J^k Y^i{}_j \tag{75}$$

We simply choose other letters like $X, Y, I, J$ to be placed on the right side of $\varphi$ and $\tilde{\varphi}$.

Still, there is an ordering ambiguity. For instance between two words:

$$|\tilde{\varphi}\varphi| IJY \quad vs \quad |\tilde{\varphi}\varphi| JIY \tag{76}$$

We simply choose an alphabetical order to arrange letters. In other words, we use the commutation relations until the letters in the word has a alphabetical order. When the word has an alphabetical order, we contract the gauge indices to form a single-trace word, and omit the gauge indices. For instance,

$$\begin{aligned}
(\tilde{\varphi}\varphi) :=& |\tilde{\varphi}^j \varphi_j| \\
(IY\tilde{\varphi})(\varphi J) :=& |\tilde{\varphi}^j \varphi_l| I_k J^l Y^k{}_j \\
(I\tilde{\varphi})(\varphi J)(IJ) :=& |\tilde{\varphi}^j \varphi_k| I_j J^k I_i J^i
\end{aligned} \tag{77}$$

As a consequence, some more steps are needed for the following:

$$|\tilde{\varphi}^j \varphi_k| I_i I_j J^k J^i \tag{78}$$

That is, we need to commute $I_i$ through $J^k$ to contract with $J^i$. While doing this, we necessarily use $[I_i, J^k] = \epsilon_1 \delta^k_i + J^k I_i$, which produces two terms.

Having fixed the ordering ambiguity, there is a few things to keep in mind additionally:

- We use F-term relation and the basic commutation relation between $X$ and $Y$ in maximum times to get rid of X's in the word, since the module only consists of $\varphi$, $\tilde{\varphi}$, $I$, $J$, $Y$.

- To use F-term relation, we first need to pull the target XY(or YX) pair to the right end, not to ruin the gauge invariance, and pull it back to the original position in the word.

- To use the superpotential relations($X\varphi = \epsilon_1 \partial_{\tilde{\varphi}}$ or $X\tilde{\varphi} = \epsilon_1 \partial_{\varphi}$), we need to bring $X$ right next to $\varphi$ or $\tilde{\varphi}$.

---

[16]Recall that $\varphi$, $\tilde{\varphi}$ are chiral multiplet scalars that are localized at the interface(between the line and the surface). In the $\Omega_{\epsilon_1}$-background, the interface localizes to a point. Hence, $\varphi$, $\tilde{\varphi}$ are localized to be at a point on the line.

Given the prescription, we would like to find $a \in \mathcal{A}_{\epsilon_1, \epsilon_2}$ and $m \in \mathcal{M}_{\epsilon_1, \epsilon_2}$ such that the value of $[a, m]$ contains $\mathcal{O}(\epsilon_1)$ terms. To illustrate the prescription, let us consider following simple example, which will not produce $\mathcal{O}(\epsilon_1)$ term.

**Example:** $[IXJ, (IY\tilde{\varphi})(\varphi J)]$

It is much clear and convenient to use closed word version for the algebra element. We will recover the open word at the end by simply multiplying $\epsilon_2$ on the closed words.

$$[TrX, (IY\tilde{\varphi})(\varphi J)] = (X) \cdot (IY\tilde{\varphi})(\varphi J) - (IY\tilde{\varphi})(\varphi J) \cdot (X) \tag{79}$$

Compute the first term:

$$
\begin{aligned}
X_0^0 |\tilde{\varphi}^b \varphi_c| I_a Y_b^a J^c &= |\tilde{\varphi}^b \varphi| I_a (\epsilon_1 \delta_b^a + Y_b^a X_0^0) J^c \\
&= \epsilon_1 |\tilde{\varphi}^b \varphi_c| I_b J^c + (IY\tilde{\varphi})(\varphi J) \cdot (X)
\end{aligned}
\tag{80}
$$

So,

$$
\begin{aligned}
[TrX, (IY\tilde{\varphi})(\varphi J)] &= \epsilon_1 |\tilde{\varphi}^b \varphi_c| I_b J^c \\
&= \epsilon_1 (I\tilde{\varphi})(\varphi J)
\end{aligned}
\tag{81}
$$

After normalization, by multiplying $\frac{\epsilon_2}{\epsilon_1^3}$ both sides, we get

$$[T[1, 0], b[z]c[1]] = \epsilon_2 b[1]c[1] \tag{82}$$

There is no $\mathcal{O}(\epsilon_1)$ correction. So, we need to work harder.

The first bi-module commutator that has an $\epsilon_1$ correction with some non-trivial dependence on $\epsilon_2$ is $\big[TrS(X^2Y), (IY\tilde{\varphi})(\varphi J)\big]$. After properly normalizing it, we have

$$
\begin{aligned}
[T[2, 1], b[z]c[1]] =& \left( -\frac{5}{3}\epsilon_2 T[0, 1] + \epsilon_2^2 b[1]c[1] \right) \\
&+ \boxed{\epsilon_1 \left( -\epsilon_2 b[1]c[1]T[0, 0] + \frac{4}{3}\epsilon_2 b[1]c[1] \right)} \\
&+ \epsilon_1^2 \left( -\frac{4}{3} b[1]c[1]T[0, 0] \right) \\
&+ \epsilon_1^3 \left( -\frac{1}{3} b[1]c[1]b[1]c[1] \right)
\end{aligned}
\tag{83}
$$

Here, we used the re-scaled basis $T[m, n]$ for $\mathcal{A}_{\epsilon_1, \epsilon_2}$. This is a better choice to be coherent with the form of the bi-module elements, since $b[z^n] = IY^n\tilde{\varphi}$ and $c[z^n] = \varphi Y^n J$ explicitly depend on $I$ and $J$. [17]We have shown the proof in Appendix §A.2.

# 4 Perturbative calculations in 5d $U(1)$ CS theory coupled to 1d QM

In this section, we will provide a derivation of the $G = U(N)$, $\hat{G} = U(1)$ ADHM algebra $\mathcal{A}_{\epsilon_1, \epsilon_2}$ using the perturbative calculation in 5d $U(1)$ CS. We will see the result from the

---

[17]Similar to the algebra case, there might be a shift in parameters $\epsilon_1$ and $\epsilon_2$ in 5d CS side; here, we simply assumed that there is no shift: $(\epsilon_1)_{5d} = (\epsilon_1)_{1d-2d}$, $(\epsilon_2)_{5d} = (\epsilon_2)_{1d-2d}$. If there were a shift in the $\epsilon_2$ dictionary, the tree level term may be a potential problem.

perturbative calculation matches with the expectation (73). The strategy, which we will spell out in detail in this section, is to compute the $\mathcal{O}(\epsilon_1{}^1)$ order gauge anomaly of various Feynman diagrams in the presence of the line defect from $M2$ brane($\mathbb{R}^1 \times \{0\} \subset \mathbb{R}^1 \times \mathbb{C}_{NC}^2$). Imposing a cancellation of the anomaly for the 5d CS theory uniquely fixes the algebra commutation relations.

Purely working in the weakly coupled 5d CS theory, we will derive the representative commutation relations of the ADHM algebra (73):

- Algebra commutation relation

$$[t[2,1], t[1,2]] = \ldots + \epsilon_1 \epsilon_2^2 t[0,0] t[0,0] + \ldots \tag{84}$$

where $t[n, m]$ is a basis element of $\mathcal{A}_{\epsilon_1, \epsilon_2}$.

As we commented in §3.1, the algebra basis used in the Feynman diagram computation is $T[m, n]$, which is related to $t[m, n]$ by rescaling with $\epsilon_2$. The effect of the change of basis is trivial in (84), so we will interchangeably use $t[m, n]$ and $T[m, n]$ without loss of generality.

## 4.1 Ingredients of Feynman diagrams

To set-up the Feynman diagram computations, we recall the 5d $U(1)$ Chern-Simons theory action on $\mathbb{R}_t \times \mathbb{C}_{NC}^2$.

$$S = \frac{1}{\epsilon_1} \int_{\mathbb{R}_t \times \mathbb{C}_{NC}^2} dz_1 dz_2 \left( A \star_{\epsilon_2} dA + \frac{2}{3} A \star_{\epsilon_2} A \star_{\epsilon_2} A \right) \tag{85}$$

with $|\epsilon_1| \ll |\epsilon_2| \ll 1$. In components, the 5d gauge field $A$ can be written as

$$A = A_t dt + A_{\bar{z}_1} d\bar{z}_1 + A_{\bar{z}_2} d\bar{z}_2 \tag{86}$$

with all the components are smooth holomorphic functions on $\mathbb{R}^1 \times \mathbb{C}_{NC}^2$.

Now, we want to collect all the ingredients of the Feynman diagram computation. It is convenient to rewrite (85) as

$$S = \frac{1}{\epsilon_1} \int_{\mathbb{R}^1 \times \mathbb{C}_{NC}^2} dz_1 dz_2 \left( AdA + \frac{2}{3} A(A \star_{\epsilon_2} A) \right) \tag{87}$$

(87) is equivalent to (85) up to a total derivative. From the kinetic term of the Lagrangian, we can read off the following information:

- 5d gauge field propagator $P$ is a solution of

$$dz_1 \wedge dz_2 \wedge dP = \delta_{t=z_1=z_2=0}. \tag{88}$$

That is,

$$P(v_1, v_2) = \langle A(v_1) A(v_2) \rangle = \frac{\bar{z}_{12} d\bar{w}_{12} dt_{12} - \bar{w}_{12} d\bar{z}_{12} dt_{12} + t_{12} d\bar{z}_{12} d\bar{w}_{12}}{d_{12}^5} \tag{89}$$

where

$$v_i = (t_i, z_i, w_i), \quad d_{ij} = \sqrt{t_{ij}^2 + |z_{ij}|^2 + |w_{ij}|^2}, \quad t_{ij} = t_i - t_j \tag{90}$$

From the three point coupling in the Lagrangian, we can extract 3-point vertex. This is not immediate, as the theory is defined on non-commutative background. Different from $U(N)$ CS, where the leading contribution of the 3-point vertex was $AAA$, the leading contribution of the 3-point coupling of the $U(1)$ gauge bosons starts from $\mathcal{O}(\epsilon_2)A\partial_{z_1}A\partial_{z_2}A$. The reason is following:

$$
\begin{aligned}
&\int dz \wedge dw \wedge A \wedge (A \star_{\epsilon_2} A) \\
&= \int A \wedge ((A_t dt + A_{\bar{z}} d\bar{z} + A_{\bar{w}} d\bar{w}) \star (A_t dt + A_{\bar{z}} d\bar{z} + A_{\bar{w}} d\bar{w})) \\
&= \int dz \wedge dw \wedge A \wedge [dt \wedge d\bar{z} (A_t \star A_{\bar{z}} - A_{\bar{z}} \star A_t) + \ldots] \\
&= \int dz \wedge dw \wedge A \wedge [dt \wedge d\bar{z} (0 + 2\epsilon_2 (\partial_z A_t \partial_w A_{\bar{z}} - \partial_w A_t \partial_z A_{\bar{z}})) + \ldots] \\
&= 2\epsilon_2 \int dz \wedge dw \wedge A \wedge [dt \wedge d\bar{z}(\partial_z A_t \partial_w A_{\bar{z}} - \partial_w A_t \partial_z A_{\bar{z}})] + \mathcal{O}(\epsilon_2^2)
\end{aligned}
\tag{91}
$$

Note that for $U(N)$ case, $SU(N)$ Lie algebra factors attached to each $A$ prevents the $\mathcal{O}(\epsilon_2^0)$ term to vanish. Still, $U(1) \subset U(N)$ part of $A$ contributes as $\mathcal{O}(\epsilon_2)$, but it can be ignored, since we take $\epsilon_2 \ll 1$.

Hence, in $U(1)$ CS, the 3-point $A\partial_z A\partial_w A$ coupling contributes as

- Three-point vertex $\mathcal{I}_{3pt}$:
$$
\mathcal{I}_{3pt} = \epsilon_2 dz \wedge dw
\tag{92}
$$

Now, we are ready to introduce the line defect into the theory and study how it couples to 5d gauge fields. Classically, $t[n_1, n_2]$ couples to the mode of 5d gauge field by

$$
\int_{\mathbb{R}} t[n_1, n_2] \partial_{z_1}^{n_1} \partial_{z_2}^{n_2} A dt
\tag{93}
$$

The last ingredient of the bulk Feynman diagram computation comes from the interaction (93).

- One-point vertex $\mathcal{I}_{1pt}^A$:
$$
\mathcal{I}_{1pt}^A = \begin{cases} t[n_1, n_2]\delta_{t,z_1,z_2} & \text{if } \partial_{z_1}^{n_1}\partial_{z_2}^{n_2}A \text{ is a part of an internal propagator} \\ t[n_1, n_2]\partial_{z_1}^{n_1}\partial_{z_2}^{n_2}A & \text{if } \partial_{z_1}^{n_1}\partial_{z_2}^{n_2}A \text{ is an external leg} \end{cases}
\tag{94}
$$

Lastly, the loop counting parameter is $\epsilon_1$. Each of the propagator is proportional to $\epsilon_1$ and the internal vertex is proportional to $\epsilon_1^{-1}$. Hence, 0-loop order($\mathcal{O}(\epsilon_1^0)$) Feynman diagrams may contain the same number of internal propagators and internal vertices and 1-loop order($\mathcal{O}(\epsilon_1)$) diagrams may contain one more internal propagators than internal vertices.

Until now, we have collected all the components of the 5d perturbative computation (89), (92), (93), and (94). With these, let us see what Feynman diagrams have non-zero BRST variations and how the cancelation of BRST variations of different diagrams leads to the ADHM algebra $\mathcal{A}_{\epsilon_1, \epsilon_2}$.

## 4.2 Feynman diagram

The goal of this section is derive the $\mathcal{O}(\epsilon_1)$-term of $[t[2, 1], t[1, 2]]$ by Feynman diagrams. We interpret the commutator $[t[2, 1], t[1, 2]]$ as the following difference between two tree level diagrams

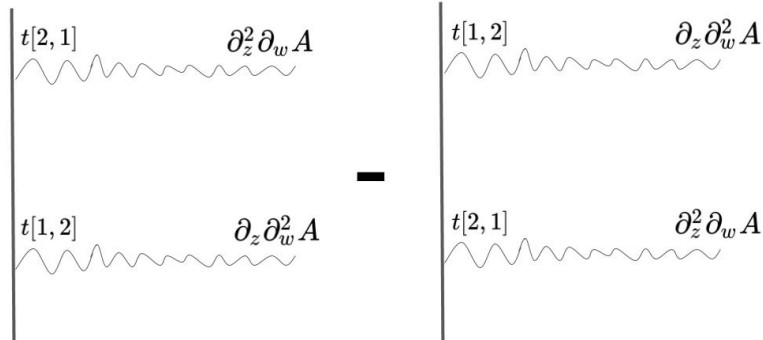

Figure 7: There is no internal propagators, but just external ghosts for 5d gauge fields, which directly interact with 1d QM. The minus sign in the middle literally means that we take a difference between two amplitudes. In the left diagram $t[1,2]$ vertex is located at $t = 0$ and $t[2,1]$ is at $t = \epsilon$. In the right diagram, $t[1,2]$ is at $t = -\epsilon$ and $t[2,1]$ at $t = 0$.

The amplitude of the diagram is

$$[t[2,1], t[1,2]] \, \partial_{z_1}^2 \partial_{z_2} A_1 \partial_{z_1} \partial_{z_2}^2 A_2 \tag{95}$$

so the BRST variation of the amplitude is proportional to

$$[t[2,1], t[1,2]] \, \partial_{z_1}^2 \partial_{z_2} A_1 \partial_{z_1} \partial_{z_2}^2 c_2 + [t[2,1], t[1,2]] \, \partial_{z_1}^2 \partial_{z_2} c_1 \partial_{z_1} \partial_{z_2}^2 A_2 \tag{96}$$

Note that the BRST variation on $A$ fields is $Q_{BRST} A = \partial c$. At $\mathcal{O}(\epsilon_1)$ level, this diagram will cancel all anomalies coming from one-loop diagrams with two external legs coupled to $\partial_{z_1}^2 \partial_{z_2} A$ and $\partial_{z_1} \partial_{z_2}^2 A$ respectively. Let's enumerate those diagrams, there are two types of diagrams:

(1) See figure 8.

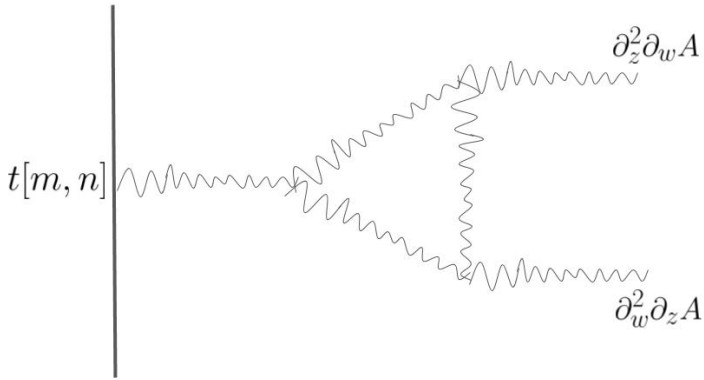

Figure 8: A diagram, which has a vanishing amplitude.

(2) See figure 9.

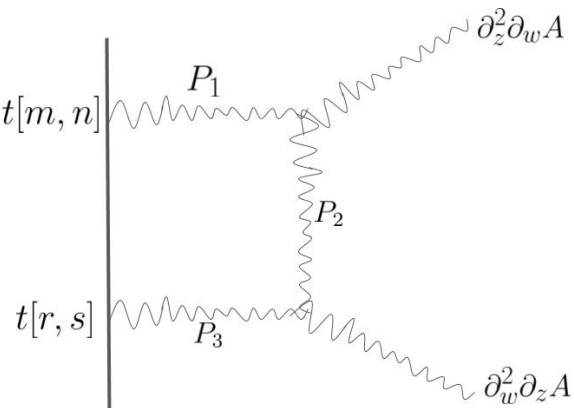

Figure 9: The vertical solid line represents the time axis, where 1d topological defect is supported. Internal wiggly lines stand for 5d gauge field propagators $P_i$, and the external wiggly lines stand for 5d gauge field $A$.

For the first diagram, we claim that the amplitude is always zero. This can be seen as follows. Let $U(1)$ act on $z$ and $w$ by rotation with weight 1, then propagators has weight $-2$. For the interaction vertex, it contains integration measure $dz \wedge dw$ together with $\partial_z$ and $\partial_w$ in the interaction term so the total weight of the interaction vertex is zero. Each external leg is of weight 3. Hence the total weight of the amplitude is $-2 - m - n < 0$, i.e. it's not invariant under the $U(1)$-rotation symmetry, so the amplitude must be zero.

For the second diagram, we will follow the approach shown in [30] and show that the diagram has a non-vanishing amplitude if and only if $m = n = r = s = 0$. And in the case that it's non-zero, it has a non-vanishing gauge anomaly consequently, under the BRST variation $Q_{BRST} A = \partial c$.

Let's do the same analysis on the second diagram as the first one, i.e. let $U(1)$ act on $z$ and $w$ by rotation with weight 1, then the total weight of the amplitude is $-n - m - r - s$. Hence the diagram is nonzero only if $m = n = r = s = 0$. In the following discussion, we will focus on he case $m = n = r = s = 0$.

We first integrate over the first vertex ($P_1 \ \partial_z^2 \partial_w A \ P_2$) and then integrate over the second vertex($P_2 \ \partial_z \partial_w^2 A \ P_3$).

**First vertex**($P_1 \ \partial_z^2 \partial_w A \ P_2$)

First, we focus on computing the integral over the first vertex:

$$\epsilon_1 \epsilon_2^2 \int_{v_1} dw_1 \wedge dz_1 \wedge \partial_{z_1} P_1(v_0, v_1) \wedge \partial_{z_2} \partial_{w_1} P_2(v_1, v_2)(z_1^2 w_1 \partial_{z_1}^2 \partial_{w_1} A) \tag{97}$$

Note that $\partial_{z_1}$ and $\partial_{w_1}$ comes from the three point coupling at $v_1$:

$$\epsilon_2 A \wedge \partial_{z_1} A \wedge \partial_{w_1} A \tag{98}$$

And $\partial_{z_2}$ comes from the 3-pt coupling at $v_2$:

$$\epsilon_2 A \wedge \partial_{z_2} A \wedge \partial_{w_2} A \tag{99}$$

We will consider $\partial_{w_2}$ later when we treat the second vertex.

The factor $z_1^2 w_1 \partial_{z_1}^2 \partial_{w_1} A$ is for the external leg attached to $v_1$, which is $c[2, 1]$. Basically, this is an ansatz, and we can start without fixing $m, n$ in $c[m, n]$. However, we will see

that the integral converges to a finite value only with this particular choice of $(m, n)$. For a simple presentation, we will drop $\partial_{z_1}^2 \partial_{w_1} A$, and recover it later.

After some manipulation, which we defer to **Lemma 1** in Appendix B.1, (97) becomes

$$- \int_{v_1} dt_1 dz_1 d\bar{z}_1 dw_1 d\bar{w}_1 \frac{|z_1|^2 |w_1|^2 \bar{z}_2 (\bar{w}_{12} dt_2 - t_{12} d\bar{w}_2)}{d_{01}^5 d_{12}^9} \tag{100}$$

The integral 100 can be further simplified by using the typical Feynman integral technique, which can be found in **Lemma 2** in Appendix B.1. We are left with

$$\bar{z}_2 (\bar{w}_2 dt_2 - t_2 d\bar{w}_2) \left( \frac{c_1}{d_{02}^5} + \frac{c_2 w_2^2}{d_{02}^7} + \frac{c_3 z_2^2}{d_{02}^7} + \frac{c_4 z_2^2 w_2^2}{d_{02}^9} \right) \tag{101}$$

with $c_i$ being a constant. Note that all the terms in the parenthesis has a same order of divergence. So, it suffices to focus on the first term to check the convergence of the full integral(we still need to do $v_2$ integral below.)

We will explicitly show the calculation for the first term, and just present the result for the second, third and fourth term in (194). They are all non-zero and finite. We will denote the first term as $\mathcal{P}$, which is 1-form.

**Second vertex($\mathcal{P} \ \partial_{z_1}^2 \partial_{z_2} A \ P_3$)**

Now, let us do the integral over the second vertex($v_2$). The remaining things are organized into

$$\int_{v_2} \mathcal{P} \wedge \partial_{w_2} P_3(v_2, v_3) \wedge dz_2 \wedge dw_2 (z_2 w_2^2 \partial_{z_2} \partial_{w_2}^2 A) \tag{102}$$

where we dropped forms related to $v_3$, as we do not integrate over it. $\partial_{w_2}$ comes from the 3-pt coupling at $v_2$:

$$\epsilon_2 A \wedge \partial_{z_2} A \wedge \partial_{w_2} A \tag{103}$$

The factor $z_2 w_2^2 \partial_{z_2} \partial_{w_2}^2 A$ is for the external leg attached to $v_2$, which corresponds to $c[1, 2]$. Again, this is an ansatz. We will see that only this integral converges and does not vanish below. We will drop $\partial_{z_2} \partial_{w_2}^2 A$ and recover it later.

The integral (102) is simplified to

$$\int_{v_2} -\frac{|z_2|^2 |w_2|^4}{d_{02}^5 d_{23}^7} dt_2 d\bar{z}_2 d\bar{w}_2 dw_2 dz_2 \tag{104}$$

The intermediate steps can be found in **Lemma 3** in Appendix B.1.

Now, it remains to evaluate the delta function at the third vertex, and use Feynman technique to evaluate the integral. By **Lemma 4** in Appendix B.1, we are left with

$$(const)\epsilon_1 \epsilon_2^2 t[0, 0] t[0, 0] \partial_{z_1}^2 \partial_{z_2} A_1 \partial_{z_1}^1 \partial_{z_2}^2 A_2 \tag{105}$$

The BRST variation of the amplitude is

$$(const)\epsilon_1 \epsilon_2^2 t[0, 0] t[0, 0] \partial_{z_1}^2 \partial_{z_2} A_1 \partial_{z_1}^1 \partial_{z_2}^2 c_2 \tag{106}$$

This indicates that the theory is quantum mechanically inconsistent, as it has a Feynman diagram that has non-zero BRST variation. However, as long as there is another diagram whose BRST variation is proportional to the same factors we can cancel the anomaly.

Hence, imposing BRST invariance of the sum of Feynman diagrams, we bootstrap the possible 1d TQM that can couple to 5d $U(1)$ CS.

An obvious choice is the tree level diagrams where $(\partial_{z_1} A)(\partial_{z_2} A)$ appears explicitly:

By equating (106) and (96), we get

$$[t[2,1], t[1,2]] = \epsilon_1 \epsilon_2^2 t[0,0] t[0,0] + \dots \tag{107}$$

So, we have reproduced the $\mathcal{O}(\epsilon_1)$ part of the ADHM algebra $\mathcal{A}_{\epsilon_1,\epsilon_2}$ commutation relation from the Feynman diagram computation:

$$[t[2,1], t[1,2]]_{\epsilon_1} = \epsilon_1 \epsilon_2^2 t[0,0] t[0,0] \tag{108}$$

where $[-,-]_{\epsilon_1}$ is the $\mathcal{O}(\epsilon_1)$-part of the commutator.

## 5  Perturbative calculations in 5d $U(1)$ CS theory coupled to 2d $\beta\gamma$

In this section, we will provide a bulk derivation of the ADHM algebra $\mathcal{A}_{\epsilon_1,\epsilon_2}$ action on the bi-module $\mathcal{M}_{\epsilon_1,\epsilon_2}$ of the ADHM algebra $\mathcal{A}_{\epsilon_1,\epsilon_2}$ using 5d Chern-Simons theory. The strategy is similar to that of the previous section. We will compute the $\mathcal{O}(\epsilon_1{}^1)$ order gauge anomaly of various Feynman diagrams in the presence of the line defect from $M2$ brane($\mathbb{R}^1 \times \{0\} \subset \mathbb{R}^1 \times \mathbb{C}^2_{NC}$), and at the same time the surface defect from $M5$ brane on ($\{0\} \times \mathbb{C} \subset \mathbb{R}^1 \times \mathbb{C}^2_{NC}$). Imposing a cancellation of the anomaly for the 5d gauge theory uniquely fixes the algebra action on the bi-module.

We will confirm the representative commutation relation between ADHM algebra and its bi-module (109) using the Feynman diagram calculation in 5d Chern-Simons, 1d topological line defect, and 2d $\beta\gamma$ coupled system.

- The algebra and the bi-module commutation relation

$$\left[t[2,1], b[z^1] c[z^0]\right]_{\epsilon_1} = \epsilon_1 \epsilon_2 \; t[0,0] c[z^0] b[z^0] + \epsilon_1 \epsilon_2 \; c[z^0] b[z^0] \tag{109}$$

where $c[z^n]$ and $b[z^m]$ are elements of the bi-module.

### 5.1  Ingredients of Feynman diagrams

The generators of the 0d bi-module $b[z^n]$, $c[z^m]$ couple to the mode of $\beta$, $\gamma$ through

$$\int_{\{0\}} \partial_{z_2}^{k_1} \beta \cdot b[z^{k_1}] + \int_{\{0\}} \partial_{z_2}^{k_2} \gamma \cdot c[z^{k_2}] \tag{110}$$

where $z = z_2$. The coupling is defined at a point, so the integral is only used for a formal presentation.

From the coupling, we learn another ingredient of the 5d-2d Feynman diagram computation:

- One-point vertices from (110):

$$
\begin{aligned}
\mathcal{I}_{1pt}^{\beta} &= \begin{cases} b[z^k]\delta_{z_2} & \text{if } \partial_{z_2}^k \beta \text{ is a part of an internal propagator} \\ b[z^k]\partial_{z_2}^k \beta & \text{if } \partial_{z_2}^k \beta \text{ is an external leg} \end{cases}, \\
\mathcal{I}_{1pt}^{\gamma} &= \begin{cases} c[z^k]\delta_{z_2} & \text{if } \partial_{z_2}^k \gamma \text{ is a part of an internal propagator} \\ c[z^k]\partial_{z_2}^k \gamma & \text{if } \partial_{z_2}^k \gamma \text{ is an external leg} \end{cases}
\end{aligned}
\tag{111}
$$

In the case of multiple $\beta, \gamma$ internal propagators flowing out, we prescribe to keep only one $\delta_{z_2}$ function.

The $\beta\gamma-$system also couples to 5d Chern-Simons theory in a canonical way:

$$\frac{1}{\epsilon_1} \int_{\mathbb{C}_{z_2}} \beta(\partial_{\bar{z}_2} - A_{\bar{z}_2} \star_{\epsilon_2})\gamma \tag{112}$$

from which we read off the last ingredients of the perturbative computation:

- The $\beta\gamma$ propagator $P_{\beta\gamma} = \langle\beta\gamma\rangle$ is a solution of

$$\partial_{\bar{z}_2} P_{\beta\gamma} = \delta_{z_2=0} \tag{113}$$

That is,

$$P_{\beta\gamma} = \langle\beta\gamma\rangle \sim \frac{dz_2}{z_2} \tag{114}$$

- The normalized three-point$(\beta, A_{5d}, \gamma)$ vertex :

$$\mathcal{I}_{3pt}^{\beta A\gamma} = 1 \tag{115}$$

Note that we are taking the lowest order vertex in the Moyal product expansion of (112), and normalize the coefficient to 1, for simplicity, in the following computation. Each $\beta\gamma$ propagator contributes $\epsilon_1$, and each $\beta A\gamma$ vertex contributes $\epsilon_1^{-1}$.

Recall that there was the gauge anomaly in the 5d CS theory in the presence of the topological line defect. Similarly, the bi-module coupled with $\beta\gamma$-system provides an additional source of the 5d gauge anomaly, since $\beta\gamma$ system has the non-trivial coupling (112) with the 5d CS theory and is charged under the 5d gauge symmetry. For the entire 5d-2d-1d coupled system to be anomaly-free, the combined gauge anomaly should be canceled. The bulk anomaly cancellation condition beautifully fixes the action of the algebra on the bi-module.

The simplest example involving the bi-module is akin to the first example of §4; notice the similarity between Fig 2 and Fig 11. As a result, the calculation in this section resembles that of §4.2.

From the ingredients provided above, we can interpret the commutator $[t[2, 1], b[z^1]c[z^0]]$ as the difference between two tree level diagrams:

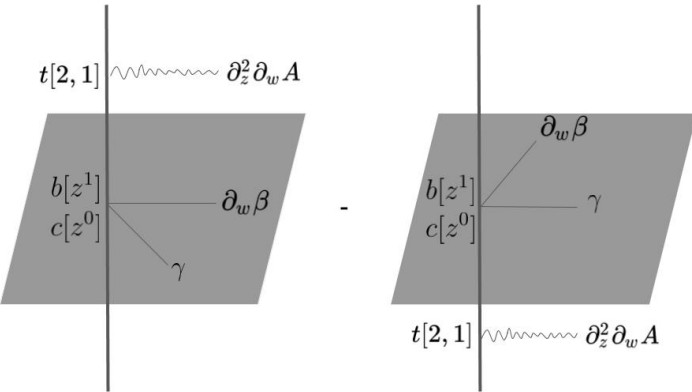

Figure 10: Feynman diagrams representing the commutator $[t[2, 1], b[z^1]c[z^0]]$. The vertical straight lines are time axis, and $\beta\gamma$ lives on the gray planes. $\beta\gamma$ only flows out of the time axis, but not flowing along the time axis. Note that there is no internal propagators of any sort. All types of lines are external legs, they are modes of $\beta, \gamma, A$.

As Fig 10 does not involve any loops, the amplitude is simply

$$[t[2, 1], b[z^1]c[z^0]] \, (\partial_z^2 \partial_w A)(\partial_w \beta)\gamma \tag{116}$$

and its BRST variation is proportional to

$$[t[2,1], b[z^1]c[z^0]] \, (\partial_z^2 \partial_w c)(\partial_w \beta)\gamma \tag{117}$$

At $\mathcal{O}(\epsilon_1)$ level, it will cancel the anomalies coming from all possible one-loop Feynman diagrams with three external legs coupled to $\partial_z^2 \partial_w A$, $\gamma$, and $\partial_w \beta$, respectively, so the only possibilities are Figure 11 and Figure 12, which we will call the diagram I and the diagram II, respectively.

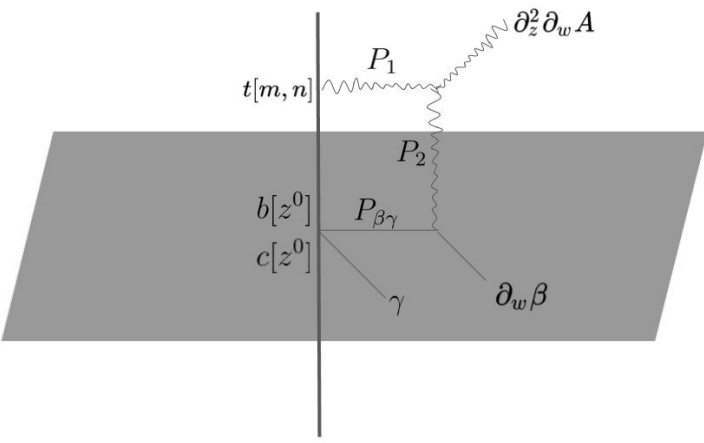

Figure 11: The Feynman diagram I. The vertical straight lines are the time axis, and the gray plane is where $\beta\gamma$-system is living. The internal horizontal straight lines are $\beta\gamma$ propagators and the external slant straight lines are modes of $\beta\gamma$. Note that no $\beta\gamma$ propagates along the time axis. The $\beta A\gamma$ three point vertex is restricted to the $\beta\gamma$-plane, but the $AAA$ three point vertex can be anywhere in the bulk.

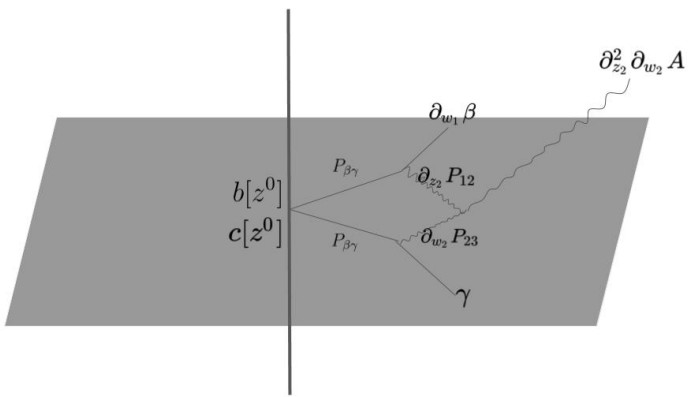

Figure 12: The Feynman diagram II.

Before we start doing concrete computations, we make a similar analysis to ADHM algebra case, i.e. let $U(1)$ rotates the $z$ and $w$ coordinates with weight 1, then $\beta - \gamma$ propagator has weight 0, Chern-Simons propagator has weight $-2$ and all interaction vertices have weight zero. It follows that the Feynman diagram I has total weight $-m-n$, and the Feynman diagram II has total weight 0. Hence the amplitudes for the Feynman diagram I is nonzero only if $m = n = 0$, so in the later discussions we will impose the condition that $m = n = 0$.

## 5.2 Feynman diagram I

In this subsection, we will show that the amplitude for Fig 11 is

$$(const) \; \epsilon_1 \; \partial_z^2 \partial_w A \partial_z \beta \; \gamma \; c[z^0] b[z^0] t[0,0] \tag{118}$$

The factor $z_2^2 w_2 \partial_{z_2}^2 \partial_{w_2} A$ is for the external leg attached to the top 3-point vertex, $v_2$. The factor corresponds to $t[2,1]$. For the convenience of presentation, We will drop $\partial_{z_2}^2 \partial_{w_2} A$ and recover it in the final result.

Along the way, we also show that the constant factor in front of (118) is finite only if the external legs are $\partial_z^2 \partial_w A \partial_z \beta \gamma$. For simplicity, we will abbreviate the leg factors during the computation.

**First vertex**

First, we focus on computing the integral over the first vertex:

$$\int_{v_1} \partial_{z_1} P_1(v_0, v_1) \wedge (w_1 dw_1) \wedge (z_1^2 dz_1) \wedge \partial_{w_1} P_2(v_1, v_2) \tag{119}$$

Note that $\partial_{z_1}$ and $\partial_{w_1}$ comes from the three point coupling at $v_1$:

$$\epsilon_2 A \wedge \partial_{z_1} A \wedge \partial_{w_1} A \tag{120}$$

In **Lemma 5** in Appendix B.2, we showed how to evaluate (119) and arrive at following expression.

$$-\int_0^1 dx \sqrt{x(1-x)}^7 \int_{v_1} [dV_1] \frac{(|z_1|^2 + x^2|z_2|^2)^2(|w_1|^2 + x^2|w_2|^2)t_2 d\bar{w}_2}{(|z_1|^2 + |w_1|^2 + t_1^2 + x(1-x)(|z_2|^2 + |w_2|^2 + t_2^2))^7} \tag{121}$$

where $[dV_1]$ is an integral measure for $v_1$ integral. We see from (121) that it was necessary to choose $c[m,n]$, $\beta_n$ to be $c[2,1]$, $\beta_1$. Otherwise, the numerator of (121) would contain holomorphic or anti-holomorphic dependence on $z_1$ or $w_1$, and this makes the $z_1$ or $w_1$ integral to vanish.

Also, we can drop terms proportional to $|z_2|^2$, since there is a delta function at the second vertex that evaluates $z_2 = 0$. So, (121) simplifies to

$$-\int_0^1 dx \sqrt{x(1-x)}^7 \int_{v_1} [dV_1] \frac{|z_1|^4(|w_1|^2 + x^2|w_2|^2)t_2 d\bar{w}_2}{(|z_1|^2 + |w_1|^2 + t_1^2 + x(1-x)(|z_2|^2 + |w_2|^2 + t_2^2))^7} \tag{122}$$

This is evaluated to

$$\frac{c_1 t_2}{d_{02}^3} + \frac{c_2 t_2 |w_2|^2}{d_{02}^5} \tag{123}$$

where $c_1$ and $c_2$ are 1-forms of $v_2$. Let us call them as $\mathcal{P}_{02}^1$ and $\mathcal{P}_{02}^2$ respectively.

**Second vertex**

Now, compute the second vertex integral, using the above computation:

$$\int_{v_2} (\mathcal{P}_{02}^1 + \mathcal{P}_{02}^2) \wedge dw_2 \frac{1}{w_2}(w_2)\delta(z_2 = 0, t_2 = \epsilon)$$

$$= \epsilon_1 \int \left( \frac{c_1}{r^5} + \frac{c_2}{r^3} \right) r dr d\theta \tag{124}$$

$$= 4\pi^4 \epsilon_1 \left( \frac{1}{43200|\epsilon|} + \frac{1}{57600|\epsilon|^3} \right)$$

We can re-scale $\epsilon$ to be 1, so the integral converges. Reinstating Gamma function factors, we finally obtain

$$(const) = \frac{\Gamma(7)}{\Gamma(7/2)\Gamma(7/2)} 4\pi^4 \left( \frac{1}{43200} + \frac{1}{57600} \right) = \frac{112\pi}{3375} \tag{125}$$

Hence, the amplitude for the Feynman diagram is

$$(const)\epsilon_1\epsilon_2 t[0,0]b[z^0]c[z^0](\partial_z^2\partial_w A)(\partial_w\beta)\gamma \tag{126}$$

Its BRST variation is

$$(const)\epsilon_1\epsilon_2 t[0,0]b[z^0]c[z^0](\partial_z^2\partial_w c)(\partial_w\beta)\gamma \tag{127}$$

By equating (127) and (117), we reproduce from the 5d gauge theory (with $\beta\gamma$-system) calculation part of the algebra action on the bi-module, which is

$$\left[t[2,1], b[z^1]c[z^0]\right] = \epsilon_1\epsilon_2 t[0,0]b[z^0]c[z^0] + \dots \tag{128}$$

## 5.3   Feynman diagram II

In this subsection we will reproduce the remaining $\mathcal{O}(\epsilon_1)$-term in (109)

$$\left[t[2,1], b[z^1]c[z^0]\right]_{\epsilon_1} = \dots + \epsilon_1\epsilon_2 b[z^0]c[z^0] + \dots \tag{129}$$

using the Feynman diagram II, see Figure 12.

   The amplitude for the diagram is

$$(const)\epsilon_2\epsilon_1 b[z^0]c[z^0] \tag{130}$$

since there are 4 internal propagators($\epsilon_1^4$) and 3 internal vertices($\epsilon_1^{-3}$), one of which is $A\partial A\partial A$ type vertex($\epsilon_2$). We will explicitly show that $(const)$ does not vanish and hence the diagram has non-zero BRST variation, which completes the RHS of (128).

**First vertex**$(P_{\beta\gamma} \ \partial_{w_1}\beta \ \partial_{z_2}P_{12})$

First, we focus on computing the integral over the first vertex:

$$\int_{v_1} \frac{1}{w_1}(w_1 dw_1)\delta(t_1 = 0, z_1 = 0) \wedge \partial_{z_2}P_{12}(v_1, v_2) \tag{131}$$

Note that $\partial_{w_2}$ comes from the three point coupling at $v_2$:

$$\epsilon_2 A \wedge \partial_{z_2} A \wedge \partial_{w_2} A \tag{132}$$

This integral evaluates to

$$-\frac{2\pi(t_2 d\bar{z}_2 + \bar{z}_2 dt_2)\bar{z}_2}{5\sqrt{t_2^2 + |z_2|^2}^5} \tag{133}$$

We presented the details in **Lemma 6.** in Appendix B.3.

**Third vertex**$(P_{\beta\gamma} \ \gamma \ \partial_{w_2}P_{23})$

Second, we focus on computing the integral over the third vertex:

$$\int_{v_3} \frac{1}{w_3}(dw_3)\delta(t_3 = 0, z_3 = 0) \wedge \partial_{w_2}P(v_2, v_3) \tag{134}$$

Note that $\partial_{w_2}$ comes from the three point coupling at $v_2$:

$$\epsilon_2 A \wedge \partial_{z_2} A \wedge \partial_{w_2} A \tag{135}$$

This integral evaluates to

$$-(t_2 d\bar{z}_2 - \bar{z}_2 dt_2)\frac{2\pi}{15w_2^2}\left(\frac{2}{\sqrt{t_2^2+|z_2|^2}^3} - \frac{5|w_2|^2 + 2t_2^2 + 2|z_2|^2}{\sqrt{t_2^2+|z_2|^2+|w_2|^2}^5}\right) \tag{136}$$

We presented the details in **Lemma 7.** in Appendix B.3.

**Second vertex**$(\partial_{z_2} P_{12} \ \partial_{z_2}^2 \partial_{w_2} A \ \partial_{w_2} P_{23})$

Now, combine (133) and (136), and compute the second vertex integral; here $z_2^n w_2^m$ denotes the external gauge boson leg.

$$\int_{v_2} dw_2 \wedge dz_2 \wedge (t_2 d\bar{z}_2 - \bar{z}_2 dt_2) \wedge (t_2 d\bar{z}_2 + \bar{z}_2 dt_2)\bar{z}_2$$

$$\times \frac{4\pi^2 z_2^n w_2^m}{75 w_2^2 \sqrt{t_2^2+|z_2|^2}^5}\left(\frac{2}{\sqrt{t_2^2+|z_2|^2}^3} - \frac{5|w_2|^2 + 2t_2^2 + 2|z_2|^2}{\sqrt{t_2^2+|z_2|^2+|w_2|^2}^5}\right)$$

$$= \int_{v_2} dw_2 \wedge dz_2 \wedge d\bar{z}_2 \wedge dt_2 \frac{4\pi^2 t_2 |z_2|^2}{75 w_2 \sqrt{t_2^2+|z_2|^2}^5}\left(\frac{2}{\sqrt{t_2^2+|z_2|^2}^3} - \frac{5|w_2|^2 + 2t_2^2 + 2|z_2|^2}{\sqrt{t_2^2+|z_2|^2+|w_2|^2}^5}\right) \tag{137}$$

We inserted $(n, m) = (2, 1)$ for the external gauge boson leg. Then, $z_2^2$ pairs with $\bar{z}_2^2$, and $w_2$ combines with $1/w_2^2$ to yield $1/w_2$. Since we do not have $d\bar{w}_2$, the integral is holomorphic integral. If $(n, m)$ were other values, the integral will simply vanish.

In **Lemma 8.** in Appendix B.3, we show that (137) is convergent, and bounded as

$$c_1 < (137) < c_2 \tag{138}$$

where $c_1$, $c_2$ are some finite constants.

Hence, the amplitude for the Feynman diagram is

$$(const)\epsilon_1\epsilon_2 b[z^0]c[z^0](\partial_z^2\partial_w A)(\partial_w \beta)\gamma \tag{139}$$

Its BRST variation is therefore non-vanishing:[18]

$$(const)\epsilon_1\epsilon_2 b[z^0]c[z^0](\partial_z^2\partial_w c)(\partial_w \beta)\gamma \tag{140}$$

This completes the remaining part of the algebra-bi-module commutation relation (**??**):

$$\left[t[2,1], b[z^1]c[z^0]\right]_{\epsilon_1} = \epsilon_1\epsilon_2 t[0,0]b[z^0]c[z^0] + \epsilon_1\epsilon_2 b[z^0]c[z^0] \tag{141}$$

# 6 Conclusion

In this paper, we studied the simplest possible configurations of M2 and M5 brane in the $\Omega-$deformed and topologically twisted M-theory. In particular, we showed the operator algebra living on M2 brane acts on the operator algebra on M5 brane and computed the

---

[18]We hope there is no confusion between the ghost for the 5d gauge field $\partial_z^2\partial_w c$ and the module element $c[z^0]$.

simplest commutators. As the M2 and M5 branes are embedded in $\Omega-$deformed and topologically twisted M-theory, the field theories on the branes have twisted holographic duals in the twisted supergravity. The dual side is interestingly captured by 5d non-commutative Chern-Simons theory coupled to a topological line defect and a vertex operator algebra. By computing several Feynman diagrams and imposing BRST invariance of the coupled system, we demonstrated that the gravity dual computation can reproduce the operator algebra commutator in the field theory. Lastly, we would like to end the paper with some open questions for future research.

First of all, the derivation of 5d Chern-Simons theory as a localization of $\Omega-$deformed and topologically twisted 11d supergravity is via type IIA/M-theory relation. We wonder how one can derive 5d Chern-Simons theory by a direct localization of 11d supergravity. We hope to study this point in future.

Second, the system we are considering in this work is the simplest configuration belong to the more general framework [19]. We can introduce more $M2_i$-branes on $\mathbb{R}_t \times \mathbb{C}_{\epsilon_i}$ and $M5_I$-branes on $\mathbb{C} \times \mathbb{C}_j \times \mathbb{C}_k$, where $i \in \{1, 2, 3\}$, $(j, k) \in \{(1, 2), (2, 3), (3, 1)\}$, and $I = \{1, 2, 3\}\backslash\{j, k\}$. Using the M-theory / type IIB duality, we can map the most general configuration to "GL-twisted type IIB" theory [53], where each M2-brane maps to $(1, 0), (0, 1), (1, 1)$ 1-brane, respectively, and each M5-brane maps to D3-brane whose boundary is provided by $(1, 0), (0, 1), (1, 1)$ 5-branes.

At the corner of the tri-valent vertex, so-called Y-algebra [28], which comes form D3-brane boundary degree of freedom [54, 55], lives. This Vertex Algebra is the most general version of our toy model $\beta\gamma$ system, and is labeled by three integers $N_1, N_2, N_3$, each of which is the number of D3-branes on three corners of the trivalent graph. So, in principle, one can extend our analysis related to the M5-brane into Y-algebra Vertex Algebra. The Koszul dual object of the the Vertex Algebra was called as universal bi-module $\mathcal{B}_{\epsilon_1, \epsilon_2}^{N_1, N_2, N_3}$ in [19].

Moreover, our ADHM algebra from $M2_1$-brane has its triality image at $M2_2$-brane and $M2_3$-brane. It was proposed in [19] that there is a co-product structure in $M2_i$-brane algebras in the Coulomb branch algebra language[19]. Hence, one can generalize our analysis related to the M2-brane into the most general algebra, obtained by fusion of three $M2_i$-brane algebra. This was called as universal algebra $\mathcal{A}_{\epsilon_1, \epsilon_2}^{n_1, n_2, n_3}$ in [19].

# Acknowledgements

We thank Davide Gaiotto and Kevin Costello for crucial advice and encouragement in various stages of this project, and their comments on our paper. We also thank Miroslav Rapcak for his comment on the draft. JO is grateful to Dongmin Gang, Hee-Cheol Kim, Jaewon Song for their questions during JO's seminar at APCTP. These questions helped us to shape the outline of the paper.

**Funding information**  The research of JO was supported in part by Kwanjeong Educational Foundation and by the Visiting Graduate Fellowship Program at the Perimeter Institute for Theoretical Physics and in part by the Berkeley Center of Theoretical Physics. Research at the Perimeter Institute is supported by the Government of Canada through Industry Canada and by the Province of Ontario through the Ministry of Economic Development & Innovation.

---

[19]It is equally possible to describe the M2-brane algebra in terms of Coulomb branch algebra, as the ADHM theory is a self-mirror in the sense of 3d mirror symmetry [56, 57].

# A  Algebra and bi-module computation

In this appendix we derive some of the commutation relations for the algebra $\mathcal{A}_{\epsilon_1,\epsilon_2}$ and the bi-module $\mathcal{M}_{\epsilon_1,\epsilon_2}$.

## A.1  Algebra

In this subsection we will take a closer look at the algebra $\mathcal{A}_{\epsilon_1,\epsilon_2}$. We begin with a formal definition of the truncated version of the algebra: the $\mathbb{C}[\epsilon_1^{\pm},\epsilon_2]$-algebra $\mathcal{A}^{(N)}$ is generated by $\{X_j^i, Y_j^i, I_i, J^j | 1 \le i, j \le N\}$ with relations

$$\overleftarrow{[X,Y]}_j^i + I_j J^i = \epsilon_2 \delta_j^i \ , \ [X_j^i, Y_l^k] = \epsilon_1 \delta_l^i \delta_j^k \ , \ [J^j, I_i] = \epsilon_1 \delta_i^j \ , \ [X_j^i, X_l^k] = [Y_j^i, Y_l^k] = 0$$
$$I_i J^j S(X^n Y^m)_j^i = (IS(X^n Y^m)J)$$
(142)

where $\overleftarrow{f(X,Y,I,J)}$ means rearranging the expression $f(X,Y,I,J)$ in the order $I < J < X < Y$, $(\cdots)$ means fully contracting all indices, the symbol $S$ means symmetrization. Similarly we define $\overrightarrow{f(X,Y,I,J)}$ as rearranging the expression $f(X,Y,I,J)$ in the order $Y < X < I < J$. The ADHM algebra $\mathcal{A}_{\epsilon_1,\epsilon_2}$ is the large $N$ limit of $\mathcal{A}^{(N)}$ where the limit is taken in the sense of the procedure in section 2.6. The first relation is the F-term relation, and the following lemma is an obvious consequence of the F-term relation:

**Lemma 1.**

$$(IS(X^n Y^m)J) = \epsilon_2(S(X^n Y^m))$$
(143)

From now on we will use $t_{n,m}$ to denote $(S(X^n Y^m))/\epsilon_1$, note that these generators are denoted by $t[n,m]$ in the rest of this paper, but here we use the subscript to make the presentation more compact. The following is clear

**Lemma 2.**

$$[t_{0,0}, t_{n,m}] = 0 \ , \ [t_{1,0}, t_{n,m}] = m t_{n,m-1} \ , \ [t_{0,1}, t_{n,m}] = n t_{n-1,m}$$
$$[t_{2,0}, t_{n,m}] = 2m t_{n+1,m-1} \ , \ [t_{1,1}, t_{n,m}] = (m-n) t_{n,m} \ , \ [t_{0,2}, t_{n,m}] = -2n t_{n-1,m+1}$$
(144)

This means that $t_{0,0}$ is central, the linear span of $t_{2,0}, t_{1,1}, t_{0,2}$ is isomorphic to $\mathfrak{sl}_2$, and the linear span of $t_{m,n}$ with $m+n = L$ is a representation of $\mathfrak{sl}_2$ of spin $L/2$.

**Lemma 3.**

$$\overleftarrow{[X,Y^n]}_j^i = n\epsilon_2 (Y^{n-1})_j^i - \sum_{a+b=n-1} \overleftarrow{(IY^a)_j (Y^b J)^i}$$
(145)

$$\overrightarrow{[X,Y^n]}_j^i = n\epsilon_2 (Y^{n-1})_j^i - \sum_{a+b=n-1} \overrightarrow{(IY^a)_j (Y^b J)^i}$$
(146)

*In particular* $\overrightarrow{[X,Y]}_j^i = \overleftarrow{[X,Y]}_j^i$.

**Lemma 4.**

$$(Y^n X Y^m)_j^i - \overleftarrow{(Y^n X Y^m)}_j^i = -\epsilon_1 \sum_{a+b=n-1} (Y^{a+m})_j^i (Y^b)$$
(147)

$$(Y^n X Y^m)_j^i - \overrightarrow{(Y^n X Y^m)}_j^i = \epsilon_1 \sum_{a+b=m-1} (Y^a)(Y^{b+n})_j^i$$
(148)

Combine Lemma 3 and 4, we immediately see that

$$
\begin{aligned}
[X, Y^n]^i_j &= \overleftarrow{[X, Y^n]^i_j} - (Y^n X)^i_j + \overleftarrow{(Y^n X)^i_j} \\
&= n\epsilon_2 (Y^{n-1})^i_j - \sum_{a+b=n-1} \overleftarrow{(IY^a)_j (Y^b J)^i} + \epsilon_1 \sum_{a+b=n-1} (Y^a)^i_j (Y^b) \\
&= n\epsilon_2 (Y^{n-1})^i_j - \sum_{a+b=n-1} \overrightarrow{(IY^a)_j (Y^b J)^i} + \epsilon_1 \sum_{a+b=n-1} (Y^a)^i_j (Y^b)
\end{aligned}
\tag{149}
$$

**Proposition 1.**

$$
\overleftarrow{(IY^a)_j (Y^b J)^i} = \overrightarrow{(IY^a)_j (Y^b J)^i} + \epsilon_1 (Y^a)^i_j (Y^b) - \epsilon_1 (Y^a)(Y^b)^i_j
\tag{150}
$$

*Proof.*

$$
\begin{aligned}
\overleftarrow{(IY^a)_j (Y^b J)^i} - \overrightarrow{(IY^a)_j (Y^b J)^i} &= I_l J^m (Y^a)^l_j (Y^b)^i_m - (Y^a)^l_j (Y^b)^i_m I_l J^m \\
&= [I_l J^m, (Y^a)^l_j (Y^b)^i_m] \\
&= [-\overleftarrow{[X, Y]^m_l}, (Y^a)^l_j (Y^b)^i_m] \\
&= (Y^a)^l_j (Y^b)^i_m \overrightarrow{[X, Y]^m_l} - \overleftarrow{[X, Y]^m_l} (Y^a)^l_j (Y^b)^i_m \\
&= \overrightarrow{(Y^b [X, Y] Y^a)^i_j} - \overleftarrow{(Y^b [X, Y] Y^a)^i_j}
\end{aligned}
$$

where in the third line we used the F-term relation and in the fourth line we used the equation $\overrightarrow{[X, Y]^m_l} = \overleftarrow{[X, Y]^m_l}$ (cf. Lemma 3). Then the result follows from Lemma 4. $\square$

**Proposition 2.**

$$
(Y^c)^i_k \overleftarrow{(IY^a)_j (Y^b J)^k} = \overleftarrow{(IY^a)_j (Y^{b+c} J)^i} + \epsilon_1 (Y^{a+c})^i_j (Y^b) - \epsilon_1 (Y^a)^i_j (Y^{b+c})
\tag{151}
$$

$$
\overrightarrow{(IY^a)_k (Y^b J)^i} (Y^c)^k_j = \overrightarrow{(IY^{a+c})_j (Y^b J)^i} + \epsilon_1 (Y^a)(Y^{b+c})^i_j - \epsilon_1 (Y^{a+c})(Y^b)^i_j
\tag{152}
$$

*Proof.*

$$
\begin{aligned}
(Y^c)^i_k \overleftarrow{(IY^a)_j (Y^b J)^k} &= (Y^c)^i_k \left( \overrightarrow{(IY^a)_j (Y^b J)^k} + \epsilon_1 (Y^a)^k_j (Y^b) - \epsilon_1 (Y^a)(Y^b)^k_j \right) \\
&= \overrightarrow{(IY^a)_j (Y^{b+c} J)^i} + \epsilon_1 (Y^{a+c})^k_j (Y^b) - \epsilon_1 (Y^a)(Y^{b+c})^k_j \\
&= \overleftarrow{(IY^a)_j (Y^{b+c} J)^i} + \epsilon_1 (Y^{a+c})^i_j (Y^b) - \epsilon_1 (Y^a)^i_j (Y^{b+c})
\end{aligned}
$$

where we used Proposition 1 to move the direction of arrows back and forth. $\square$

**Proposition 3.**

$$
\frac{(XY^m)}{\epsilon_1} = t_{1,m} + \frac{1}{m+1} \sum_{k=0}^{m-1} (k+1)(Y^k)(Y^{m-1-k})
\tag{153}
$$

$$
\frac{(Y^m X)}{\epsilon_1} = t_{1,m} - \frac{1}{m+1} \sum_{k=0}^{m-1} (k+1)(Y^k)(Y^{m-1-k})
\tag{154}
$$

*Proof.*

$$
\begin{aligned}
(m+1)\frac{(XY^m)}{\epsilon_1} - (m+1)t_{1,m} &= \frac{1}{\epsilon_1} \sum_{a+b=m} ([X, Y^a] Y^b) \\
&= \sum_{r+s+t=m-1} (Y^r)(Y^{s+t})
\end{aligned}
$$

Similar for the other one. $\square$

**The Key Commutation Relation**

There is a $SL_2$-symmetry on the algebra $\mathcal{A}^{(N)}$ under which $(X, Y)$ transforms as a vector. We will use the following particular transform

$$\phi_\alpha : X \mapsto X \ , \ Y \mapsto Y + \alpha X$$

where $\alpha$ is a formal parameter. Consider

$$A := \sum_{a+b=n-1} (([Y^a, X]Y^b X) + (XY^a[X, Y^b])) = \frac{\mathrm{d}}{\mathrm{d}\alpha}\phi_\alpha \left((XY^n) + (Y^n X)\right) - 2n(XY^{n-1}X) \tag{155}$$

This leads to

$$3A + 2n([X, Y^{n-1}X]) + 2n([XY^{n-1}, X]) = 3\frac{\mathrm{d}}{\mathrm{d}\alpha}\phi_\alpha \left((XY^n) + (Y^n X)\right) - 2\epsilon_1[t_{3,0}, t_{0,n}]$$
$$= 6n\epsilon_1 t_{2,n-1} - 2\epsilon_1[t_{3,0}, t_{0,n}] \tag{156}$$

It follows that

$$= 3nt_{2,n-1} - \frac{3A}{2\epsilon_1} - \frac{n}{\epsilon_1}([X, Y^{n-1}X]) - \frac{n}{\epsilon_1}([XY^{n-1}, X])$$
$$= 3nt_{2,n-1} - \frac{3A}{2\epsilon_1} + n \sum_{a+b=n-2} \left((XY^a)(Y^b) - (Y^a)(Y^b X)\right)$$
$$= 3nt_{2,n-1} - \frac{3A}{2\epsilon_1} + n \sum_{a+b=n-2} [(XY^a), (Y^b)] + 2n\epsilon_1 \sum_{u+v+w=n-3} \frac{u+1}{u+v+2}(Y^u)(Y^v)(Y^w)$$
$$= 3nt_{2,n-1} - \frac{3A}{2\epsilon_1} + \epsilon_1^2 \frac{n(n-1)(n-2)}{2}t_{0,n-3} + n\epsilon_1 \sum_{u+v+w=n-3} (Y^u)(Y^v)(Y^w) \tag{157}$$

We have following assertion which will be proven in the end of this subsection

**Lemma 5.**

$$A = \epsilon_2(\epsilon_1 + \epsilon_2)\sum_{m=0}^{n-3}(m+1)(n-2+m)(Y^m)(Y^{n-3-m}) - \epsilon_2(\epsilon_1 + \epsilon_2)\binom{n}{3}(Y^{n-3})$$
$$+ \epsilon_1^2\binom{n}{3}(Y^{n-3}) + \frac{2n\epsilon_1^2}{3}\sum_{u+v+w=n-3}(Y^u)(Y^v)(Y^w) \tag{158}$$

Plug it into the equation 157 and we obtain the following

**Proposition 4** (The Key Commutation Relation)**.** *Let* $\sigma_2 = \epsilon_1^2 + \epsilon_2^2 + \epsilon_1\epsilon_2$ *and* $\sigma_3 = -\epsilon_1\epsilon_2(\epsilon_1 + \epsilon_2)$, *then*

$$[t_{3,0}, t_{0,n}] = 3nt_{2,n-1} + \frac{3\sigma_2}{2}\binom{n}{3}t_{0,n-3} + \frac{3\sigma_3}{2}\sum_{m=0}^{n-3}(m+1)(n-2+m)t_{0,m}t_{0,n-3-m} \tag{159}$$

Proposition 4 together with Lemma 2 actually determine all the other commutation rela-

tions as following: first of all we have

$$
\begin{aligned}
[t_{3,0}, t_{n,m}] &= \frac{1}{2^n m! \binom{n+m}{m}} \mathrm{ad}_{t_{2,0}}^n \left([t_{3,0}, t_{0,n+m}]\right) \\
&= 3m t_{n+2,m-1} + \frac{3\sigma_2}{2} \binom{m}{3} t_{n,m-3} \\
&\quad + \frac{3\sigma_3}{2} \sum_{b=0}^{m-3} \sum_{a=0}^{n} (a+1)(n-a+1) \frac{\binom{a+b+1}{a+1} \binom{m+n-a-b-2}{n-a+1}}{\binom{n+m}{m}} t_{a,b} t_{n-a,m-3-b}
\end{aligned}
$$

then for $a + b = 3$, $[t_{a,b}, t_{n,m}]$ is obtained by applying $\mathrm{ad}_{t_{0,2}}$ to $[t_{3,0}, t_{n',m'}]$. Suppose that $[t_{a,b}, t_{n,m}]$ is obtained for all $a + b \leq k$ and all pairs $(n, m)$, then $[t_{k+1,0}, t_{n,m}]$ can be obtained by applying $\mathrm{ad}_{t_{3,0}}$ to $[t_{k-1,1}, t_{n,m}]$. Hence the general $[t_{a,b}, t_{n,m}]$ is obtained by induction on $k$.

*Proof of Lemma 5.*

$$A = \sum_{a+b=n-1} \left( ([Y^a, X]Y^b X) + (XY^a[X, Y^b]) \right)$$

$$= \sum_{a+b=n-1} \left( \sum_{s+t=a-1} \overleftarrow{(IY^{s+b})_i (Y^t J)^j} X_j^i - a\epsilon_2 (Y^{n-2}X) - \epsilon_1 \sum_{s+t=a-1} (Y^t)(Y^{s+b}X) \right)$$

$$+ \sum_{a+b=n-1} \left( b\epsilon_2 (XY^{n-2}) + \epsilon_1 \sum_{s+t=b-1} (XY^s)(Y^{t+a}) - \sum_{s+t=b-1} X_j^i \overleftarrow{(IY^s)_i (Y^{t+a} J)^j} \right)$$

$$= \sum_{a+b=n-1} \left( \sum_{s+t=a-1} I_k J^l \overrightarrow{(Y^{s+b} X Y^t)_l^k} - a\epsilon_2 (Y^{n-2}X) - \epsilon_1 \sum_{s+t=a-1} (Y^t)(Y^{s+b}X) \right)$$

$$+ \sum_{a+b=n-1} \left( b\epsilon_2 (XY^{n-2}) + \epsilon_1 \sum_{s+t=b-1} (XY^{s+a})(Y^t) - \sum_{s+t=b-1} I_k J^l \overleftarrow{(Y^s X Y^{t+a})_l^k} \right)$$

$$= \epsilon_2 \binom{n}{2} ([X, Y^{n-2}]) + \epsilon_1 \sum_{r+s+t=n-2} \left( (XY^{r+s})(Y^s) - (Y^t)(Y^{r+s}X) \right)$$

$$+ \sum_{r+s+t=n-2} I_k J^l \overleftarrow{(Y^r [Y^s, X] Y^t)_l^k} - \epsilon_1 \epsilon_2 \sum_{r+s+t+u=n-3} (Y^{r+s+t})(Y^u)$$

$$- \epsilon_1 \epsilon_2 \sum_{r+s+t=n-3} (r+t+2)(Y^{r+s})(Y^t)$$

$$= \sum_{r+s+t+u=n-3} I_k J^l \overleftarrow{(IY^{r+s})_l (Y^{t+u} J)^k} - \sum_{r+s+t=n-2} s\epsilon_2 I_k J^l \overleftarrow{(Y^{r+s+t-1})_l^k}$$

$$+ \epsilon_1 \sum_{r+s+t=n-2} \left( (XY^{r+s})(Y^t) - (Y^t)(Y^{r+s}X) \right)$$

$$= \epsilon_2 (\epsilon_1 + \epsilon_2) \sum_{m=0}^{n-3} (m+1)(n-2+m)(Y^m)(Y^{n-3-m}) - \epsilon_2 (\epsilon_1 + \epsilon_2) \binom{n}{3} (Y^{n-3})$$

$$+ \epsilon_1 \sum_{r+s+t=n-2} \left( (XY^{r+s})(Y^t) - (Y^t)(Y^{r+s}X) \right)$$

$$= \epsilon_2 (\epsilon_1 + \epsilon_2) \sum_{m=0}^{n-3} (m+1)(n-2+m)(Y^m)(Y^{n-3-m}) - \epsilon_2 (\epsilon_1 + \epsilon_2) \binom{n}{3} (Y^{n-3})$$

$$+ \epsilon_1^2 \sum_{r+s+t=n-2} [t_{1,r+s}, (Y^t)] + 2\epsilon_1^2 \sum_{u+v+w=n-3} (u+1)(Y^u)(Y^v)(Y^w)$$

$$= \epsilon_2 (\epsilon_1 + \epsilon_2) \sum_{m=0}^{n-3} (m+1)(n-2+m)(Y^m)(Y^{n-3-m}) - \epsilon_2 (\epsilon_1 + \epsilon_2) \binom{n}{3} (Y^{n-3})$$

$$+ \epsilon_1^2 \binom{n}{3} (Y^{n-3}) + \frac{2n\epsilon_1^2}{3} \sum_{u+v+w=n-3} (Y^u)(Y^v)(Y^w)$$

$$\tag{160}$$

Some explanation: from 5th equality to 6th equality, the essential computation is the

following:

$$
\begin{aligned}
\sum_{r+s+t+u=n-3} I_k J^l \overleftarrow{(IY^{r+s})_l (Y^{t+u}J)^k} &= \sum_{r+s+t+u=n-3} I_k J^l I_i J^j (Y^{r+s})^i_l (Y^{t+u})^k_j \\
&= \sum_{r+s+t+u=n-3} I_k I_i J^j J^l (Y^{r+s})^i_l (Y^{t+u})^k_j + \epsilon_1 \sum_{r+s+t+u=n-3} I_k J^j \delta^l_i (Y^{r+s})^i_l (Y^{t+u})^k_j \\
&= \sum_{r+s+t+u=n-3} I_k J^j I_i J^l (Y^{r+s})^i_l (Y^{t+u})^k_j - \epsilon_1 \sum_{r+s+t+u=n-3} I_k J^l \delta^j_i (Y^{r+s})^i_l (Y^{t+u})^k_j \\
&+ \epsilon_1 \sum_{r+s+t+u=n-3} I_k J^j \delta^l_i (Y^{r+s})^i_l (Y^{t+u})^k_j \\
&= \epsilon_2 (\epsilon_1 + \epsilon_2) \sum_{r+s+t+u=n-3} (Y^{r+s})(Y^{t+u}) - \epsilon_2 \epsilon_1 \binom{n}{3} (Y^{n-3})
\end{aligned}
\tag{161}
$$

Then we define $m = r + s$, then there are $m + 1$ ways of decomposing $m$ as $r + s$, similarly there are $n - 3 - m + 1$ ways of decomposing $n - 3 - m$ as $t + u$, hence the result can be simplified to

$$
\epsilon_2 (\epsilon_1 + \epsilon_2) \sum_{m=0}^{n-3} (m+1)(n-2+m)(Y^m)(Y^{n-3-m}) - \epsilon_2 \epsilon_1 \binom{n}{3} (Y^{n-3})
\tag{162}
$$

$\square$

## A.2   Bi-module

The simplest algebra, bi-module commutator that has $\epsilon_1$ correction in the RHS is

$$
\begin{aligned}
[T[2,1], b[z]c[1]] =& \left( -\frac{5}{3}\epsilon_2 T[0,1] + \epsilon_2^2 b[1]c[1] \right) \\
&+ \epsilon_1 \left( -\epsilon_2 b[1]c[1]T[0,0] + \frac{4}{3}\epsilon_2 b[1]c[1] \right) \\
&+ \epsilon_1^2 \left( -\frac{4}{3} b[1]c[1]T[0,0] \right) \\
&+ \epsilon_1^3 \left( -\frac{1}{3} b[1]c[1]b[1]c[1] \right)
\end{aligned}
\tag{163}
$$

We will prove it in this section.

Let us expand the LHS.

$$
\begin{aligned}
\big[ S(X^2 Y), (IY\tilde{\varphi})(\varphi J) \big] =& \frac{1}{3}(XXY + XYX + YXX)\cdot(IY\tilde{\varphi})(\varphi J) \\
&- \frac{1}{3}(IY\tilde{\varphi})(\varphi J)\cdot(XXY + XYX + YXX)
\end{aligned}
\tag{164}
$$

Compute the first term:

$$
\begin{aligned}
(XXY) \cdot (IY\tilde{\varphi})(\varphi J) &= X_1^0 X_2^1 |\tilde{\varphi}^b \varphi_c| I_a Y_b^a J^c Y_0^2 + X_1^0 X_2^1 |\tilde{\varphi}^b \varphi_c \tilde{\varphi}^2 \varphi_0| I_a Y_b^a J^c \\
&= |\tilde{\varphi}^b \varphi_c| I_a X_1^0 (\epsilon_1 \delta_b^1 \delta_2^a + Y_b^a X_2^1) J^c Y_0^2 + \epsilon_1 X_1^0 |\tilde{\varphi}^b (\delta_c^1 \varphi_0 + \delta_0^1 \varphi_c)| I_a Y_b^a J^c \\
&= \epsilon_1 |\tilde{\varphi}^b \varphi_c| I_2 X_b^0 J^c Y_0^2 + \epsilon_1 |\tilde{\varphi}^b \varphi_c| I_a (\epsilon_1 \delta_b^0 \delta_1^a + Y_b^a X_1^0) X_2^1 J^c Y_0^2 + \epsilon_1 |\tilde{\varphi}^b \varphi_0| I_a X_c^0 Y_b^a J^c \\
&\quad + \epsilon_1 |\tilde{\varphi}^b \varphi_c| I_a (X) Y_b^a J^c \\
&= \epsilon_1 (-\epsilon_1)(IYJ) + \epsilon_1 |\tilde{\varphi}^0 \varphi_c| I_1 J^c X_2^1 Y_0^2 + \underline{(IY\tilde{\varphi})(\varphi J)(X^2 Y)} + (-\epsilon_1)\epsilon_1 (IYJ) \\
&\quad + \epsilon_1 |\tilde{\varphi}^b \varphi_c| I_a (\epsilon_1 \delta_b^a + Y_b^a (X)) J^c \\
&= -\epsilon_1^2 \epsilon_2(Y) + \epsilon_1 (IXY\tilde{\varphi})(\varphi J) + \underline{(IY\tilde{\varphi})(\varphi J) \cdot (XXY)} - \epsilon_1^2 \epsilon_2(Y) \\
&\quad + \epsilon_1^2 (I\tilde{\varphi})(\varphi J) + \epsilon_1 (IY\tilde{\varphi})(\varphi J)(X) \\
&= -2\epsilon_1^2 \epsilon_2(Y) + \epsilon_1 (IXY\tilde{\varphi})(\varphi J) + \underline{(IY\tilde{\varphi})(\varphi J) \cdot (XXY)} + \epsilon_1^2 (I\tilde{\varphi})(\varphi J) \\
&\quad + \epsilon_1 (IY\tilde{\varphi})(\varphi J)(X)
\end{aligned} \tag{165}
$$

So,

$$
\begin{aligned}
[(XXY),(IY\tilde{\varphi})(\varphi J)] = &-2\epsilon_1^2 \epsilon_2(Y) + \epsilon_1 (IXY\tilde{\varphi})(\varphi J) + \epsilon_1^2 (I\tilde{\varphi})(\varphi J) \\
&+ \epsilon_1 (IY\tilde{\varphi})(\varphi J)(X)
\end{aligned} \tag{166}
$$

Next,

$$
\begin{aligned}
(XYX) \cdot (IY\tilde{\varphi})(\varphi J) &= X_1^0 Y_2^1 |\tilde{\varphi}^b \varphi_c| I_a (\epsilon_1 \delta_b^2 \delta_0^a + Y_b^a X_0^2) J^c \\
&= \epsilon_1 |\tilde{\varphi}^2 \varphi_c| I_0 X_1^0 Y_2^1 J^c + \epsilon_1 |\tilde{\varphi}^2 \varphi_c \tilde{\varphi}^1 \varphi_2| I_0 X_1^0 J^c + |\tilde{\varphi}^b \varphi_c| I_a X_1^0 Y_2^1 Y_b^a X_0^2 J^c \\
&\quad + |\tilde{\varphi}^b \varphi_c \tilde{\varphi}^1 \varphi_2| I_a X_1^0 Y_b^a X_0^2 J^c \\
&= \epsilon_1 (IXY\tilde{\varphi})(\varphi J) + \epsilon_1 (-\epsilon_1)((\tilde{\varphi}\varphi)(IJ) + (I\tilde{\varphi})(\varphi J)) \\
&\quad + |\tilde{\varphi}^b \varphi_c| I_a (\epsilon_1 \delta_b^0 \delta_1^a + Y_b^a X_1^0) J^c Y_2^1 X_0^2 + (-\epsilon_1)(|\tilde{\varphi}^b \varphi_2| I_a Y_b^a X_0^2 J^0 + |\tilde{\varphi}^b \varphi_c| I_a Y_b^a J^c (X)) \\
&= \epsilon_1 (IXY\tilde{\varphi})(\varphi J) - \epsilon_1^2 (\tilde{\varphi}\varphi)(IJ) - \epsilon_1^2 (I\tilde{\varphi})(\varphi J) + \epsilon_1 |\tilde{\varphi}^0 \varphi_c| I_1 J^c Y_2^1 X_0^2 \\
&\quad + \underline{(IY\tilde{\varphi})(\varphi J)(XYX)} - \epsilon_1 |\tilde{\varphi}^b \varphi_2| I_a (-\epsilon_1 \delta_0^a \delta_b^2 + X_0^2 Y_b^a) J^0 - \epsilon_1 (IY\tilde{\varphi})(\varphi J)(X) \\
&= \epsilon_1 (IXY\tilde{\varphi})(\varphi J) - \epsilon_1^2 (\tilde{\varphi}\varphi)(IJ) - \epsilon_1^2 (I\tilde{\varphi})(\varphi J) + \epsilon_1 |\tilde{\varphi}^0 \varphi_c| I_1 J^c (-\epsilon_1 N \delta_0^1 + X_0^2 Y_2^1) \\
&\quad + \underline{(IY\tilde{\varphi})(\varphi J)(XYX)} + \epsilon_1^2 (\tilde{\varphi}\varphi)(IJ) - \epsilon_1 (-\epsilon_1)(IYJ) \\
&= \epsilon_1 (IXY\tilde{\varphi})(\varphi J) - \epsilon_1^2 (I\tilde{\varphi})(\varphi J) - \epsilon_1^2 N (I\tilde{\varphi})(\varphi J) - \epsilon_1^2 (IYJ) + \epsilon_1^2 (IYJ) \\
&\quad + \underline{(IY\tilde{\varphi})(\varphi J)(XYX)} \\
&= \epsilon_1 (IXY\tilde{\varphi})(\varphi J) - \epsilon_1^2 (I\tilde{\varphi})(\varphi J) - \epsilon_1^2 N (I\tilde{\varphi})(\varphi J) + \underline{(IY\tilde{\varphi})(\varphi J)(XYX)}
\end{aligned} \tag{167}
$$

So,

$$
[(XYX),(IY\tilde{\varphi})(\varphi J)] = \epsilon_1 (IXY\tilde{\varphi})(\varphi J) - \epsilon_1^2 (I\tilde{\varphi})(\varphi J) - \epsilon_1^2 N (I\tilde{\varphi})(\varphi J) \tag{168}
$$

Next,

$$
\begin{aligned}
(YXX)\cdot(IY\tilde{\varphi})(\varphi J) &= Y_1^0|\tilde{\varphi}^b\varphi_c|I_aX_2^1(\epsilon_1\delta_b^2\delta_0^a + Y_b^aX_0^2)J^c \\
&= \epsilon_1 Y_1^0|\tilde{\varphi}^2\varphi_c|I_0X_2^1J^c + Y_1^0|\tilde{\varphi}^b\varphi_c|I_a(\epsilon_1\delta_b^1\delta_2^a + Y_b^aX_2^1)X_0^2J^c \\
&= \epsilon_1(-\epsilon_1)(IYJ) + \epsilon_1 Y_1^0|\tilde{\varphi}^1\varphi_c|I_aX_0^aJ^c + |\tilde{\varphi}^b\varphi_c\tilde{\varphi}^0\varphi_1|I_aY_b^aX_2^1X_0^2J^c + \underline{(IY\tilde{\varphi})(\varphi J)(YXX)} \\
&= -\epsilon_1^2\epsilon_2(Y) + \epsilon_1(IXY\tilde{\varphi})(\varphi J) + \epsilon_1(-N\epsilon_1)(I\tilde{\varphi})(\varphi J) \\
&\quad + \epsilon_1|\tilde{\varphi}^1\varphi_c\varphi^0\varphi_1|I_aX_0^aJ^c + |\tilde{\varphi}^b\varphi_c\varphi^0\varphi_1|I_a(-\epsilon_1\delta_2^a\delta_b^1 + X_2^1Y_b^a)X_0^2J^c + \underline{(IY\tilde{\varphi})(\varphi J)(YXX)} \\
&= -\epsilon_1^2\epsilon_2(Y) + \epsilon_1(IXY\tilde{\varphi})(\varphi J) - N\epsilon_1^2(I\tilde{\varphi})(\varphi J) + \epsilon_1(-\epsilon_1)(\tilde{\varphi}\varphi)(I\tilde{\varphi})(\varphi J) \\
&\quad + \epsilon_1(-\epsilon_1)(\tilde{\varphi}\varphi)(IJ) - \epsilon_1|\tilde{\varphi}^1\varphi_c\tilde{\varphi}^0\varphi_1|I_2X_0^2J^c \\
&\quad + (-\epsilon_1)(|\tilde{\varphi}^b\varphi_c|I_aY_b^aJ^c(X) + |\tilde{\varphi}^0\varphi_c|I_aY_2^aX_0^2J^c) + \underline{(IY\tilde{\varphi})(\varphi J)(YXX)} \\
&= -\epsilon_1^2\epsilon_2(Y) + \epsilon_1(IXY\tilde{\varphi})(\varphi J) - N\epsilon_1^2(I\tilde{\varphi})(\varphi J) - \epsilon_1^2(\tilde{\varphi}\varphi)(I\tilde{\varphi})(\varphi J) - \epsilon_1^2(\tilde{\varphi}\varphi)(IJ) \\
&\quad - \epsilon_1(-\epsilon_1)(I\tilde{\varphi})(\varphi J) - \epsilon_1(-\epsilon_1)(\tilde{\varphi}\varphi)(IJ) - \epsilon_1(IY\tilde{\varphi})(\varphi J)(X) \\
&\quad - \epsilon_1(-\epsilon_1 N)(I\tilde{\varphi})(\varphi J) - \epsilon_1(-\epsilon_1)(IYJ) + \underline{(IY\tilde{\varphi})(\varphi J)(YXX)} \\
&= \epsilon_1(IXY\tilde{\varphi})(\varphi J) - \epsilon_1(IY\tilde{\varphi})(\varphi J)(X) + \epsilon_1^2(I\tilde{\varphi})(\varphi J) - \epsilon_1^2(\tilde{\varphi}\varphi)(I\tilde{\varphi})(\varphi J) \\
&\quad + \underline{(IY\tilde{\varphi})(\varphi J)(YXX)}
\end{aligned}
$$

$$(169)$$

So,

$$
\begin{aligned}
[(YXX),(IY\tilde{\varphi})(\varphi J)] =& \epsilon_1(IXY\tilde{\varphi})(\varphi J) - \epsilon_1(IY\tilde{\varphi})(\varphi J)(X) + \epsilon_1^2(I\tilde{\varphi})(\varphi J) \\
& - \epsilon_1^2(\tilde{\varphi}\varphi)(I\tilde{\varphi})(\varphi J)
\end{aligned}
$$

$$(170)$$

Collecting above, we have

$$
\begin{aligned}
\left[S(X^2Y),(IY\tilde{\varphi})(\varphi J)\right] =& \frac{1}{3}\Bigg( -2\epsilon_1^2\epsilon_2(Y) + \epsilon_1(IXY\tilde{\varphi})(\varphi J) + \epsilon_1^2(I\tilde{\varphi})(\varphi J) \\
& + \epsilon_1(IY\tilde{\varphi})(\varphi J)(X) + \epsilon_1(IXY\tilde{\varphi})(\varphi J) - \epsilon_1^2(I\tilde{\varphi})(\varphi J) - \epsilon_1^2 N(I\tilde{\varphi})(\varphi J) \\
& + \epsilon_1(IXY\tilde{\varphi})(\varphi J) - \epsilon_1(IY\tilde{\varphi})(\varphi J)(X) + \epsilon_1^2(I\tilde{\varphi})(\varphi J) - \epsilon_1^2(\tilde{\varphi}\varphi)(I\tilde{\varphi})(\varphi J) \Bigg) \\
=& \epsilon_1(IXY\tilde{\varphi})(\varphi J) - \frac{2}{3}\epsilon_1^2\epsilon_2(Y) - \frac{1}{3}\epsilon_1^2 N(I\tilde{\varphi})(\varphi J) - \frac{1}{3}\epsilon_1^2(\tilde{\varphi}\varphi)(I\tilde{\varphi})(\varphi J) \\
& + \frac{1}{3}\epsilon_1^2(I\tilde{\varphi})(\varphi J)
\end{aligned}
$$

$$(171)$$

We are not done yet, since $(IXY\tilde{\varphi})(\varphi J)$ is reducible by the F-term relation.

$$
\begin{aligned}
\epsilon_1|\tilde{\varphi}^0\varphi_c|I_1J^cX_2^1Y_0^2 =& \epsilon_1|\tilde{\varphi}^0\varphi_c|I_1J^c(X_0^2Y_2^1 - (I_0J^1 - \epsilon_2\delta_0^1)) \\
=& \epsilon_1(-\epsilon_1)(IYJ) - \epsilon_1|\tilde{\varphi}^0\varphi_c|(J^cI_1 - \epsilon_1\delta_1^c)I_0J^1 + \epsilon_1\epsilon_2(I\tilde{\varphi})(\varphi J) \\
=& -\epsilon_1^2(IYJ) - \epsilon_1|\tilde{\varphi}^0\varphi_c|(I_0J^c + \epsilon_1\delta_0^c)I_1J^1 + \epsilon_1^2(I\tilde{\varphi})(\varphi J) \\
& + \epsilon_1\epsilon_2(I\tilde{\varphi})(\varphi J) \\
=& -\epsilon_1^2(IYJ) - \epsilon_1(I\tilde{\varphi})(\varphi J)(IJ) - \epsilon_1^2(\tilde{\varphi}\varphi)(IJ) + \epsilon_1^2(I\tilde{\varphi})(\varphi J) \\
& + \epsilon_1\epsilon_2(I\tilde{\varphi})(\varphi J)
\end{aligned}
$$

$$(172)$$

Plugging this into (171), we get

$$\left[S(X^2Y),(IY\tilde{\varphi})(\varphi J)\right] = (-\epsilon_1^2(IYJ) - \epsilon_1(I\tilde{\varphi})(\varphi J)(IJ) - \epsilon_1^2(\tilde{\varphi}\varphi)(IJ) + \epsilon_1^2(I\tilde{\varphi})(\varphi J)$$
$$+ \epsilon_1\epsilon_2(I\tilde{\varphi})(\varphi J)) - \frac{2}{3}\epsilon_1^2\epsilon_2(Y) - \frac{1}{3}\epsilon_1^2(\tilde{\varphi}\varphi)(I\tilde{\varphi})(\varphi J)$$
$$- \frac{1}{3}\epsilon_1^2 N(I\tilde{\varphi})(\varphi J) + \frac{1}{3}\epsilon_1^2(I\tilde{\varphi})(\varphi J)$$

$$(173)$$

After normalization, by multiplying $\frac{\epsilon_2}{\epsilon_1^3}$ both sides, and using the identity[20]

$$(\tilde{\varphi}\varphi)\epsilon_2 = (I\tilde{\varphi})(\varphi J) \tag{175}$$

we have

$$[T[2,1],b[z]c[1]] = \left(-\frac{5}{3}\epsilon_2 T[0,1] + \epsilon_2^2 b[1]c[1]\right)$$
$$+ \epsilon_1\left(-\epsilon_2 b[1]c[1]T[0,0] + \frac{4}{3}\epsilon_2 b[1]c[1]\right)$$
$$+ \epsilon_1^2\left(-\frac{4}{3}b[1]c[1]T[0,0]\right)$$
$$+ \epsilon_1^3\left(-\frac{1}{3}b[1]c[1]b[1]c[1]\right)$$

$$(176)$$

# B   Intermediate steps in Feynman diagram calculations

## B.1   Intermediate steps in section 4.2

**Lemma 1.**

We will compute the following integral.

$$\epsilon_1\epsilon_2^2 \int_{v_1} dw_1 \wedge dz_1 \wedge \partial_{z_1}P_1(v_0,v_1) \wedge \partial_{z_2}\partial_{w_1}P_2(v_1,v_2)(z_1^2 w_1\partial_{z_1}^2\partial_{w_1}A) \tag{177}$$

Computing the partial derivatives, we can re-write it as

$$\epsilon_1\epsilon_2^2\left(\frac{\bar{z}_1}{d_{01}^2}\frac{\bar{w}_1}{d_{12}^4}(w_1 z_1\bar{z}_2)\right)[P(v_0,v_1)\wedge dw_1 \wedge z_1 dz_1 \wedge P(v_1,v_2)] \tag{178}$$

Note that we ignore all constant factors here. We see that

$$P(v_0,v_1)\wedge P(v_1,v_2) = \frac{d\bar{z}_1 d\bar{w}_1 dt_1}{d_{01}^5 d_{12}^5}(\bar{z}_{01}\bar{w}_{12}dt_2 - \bar{z}_{01}t_{12}d\bar{w}_2 + \bar{w}_{01}t_{12}d\bar{z}_2$$
$$- \bar{w}_{01}\bar{z}_{12}dt_2 + t_{01}\bar{z}_{12}d\bar{w}_2 - t_{01}\bar{w}_{12}d\bar{z}_2) \tag{179}$$

---

[20]The identity can be derived using the F-term relation:

$$\tilde{\varphi}^i\left([X,Y]_i^j + I_iJ^j - \epsilon_2\delta_i^j\right)\varphi_j = 0$$
$$(Y) - (Y) + (I\tilde{\varphi})(\varphi J) - \epsilon_2(\tilde{\varphi}\varphi) = 0 \tag{174}$$
$$(I\tilde{\varphi})(\varphi J) = \epsilon_2(\tilde{\varphi}\varphi)$$

Including $\wedge dw_1 \wedge (z_1 dz_1)\wedge$, we can simplify it:

$$
P(v_0, v_1) \wedge P(v_1, v_2) \wedge (w_1 dw_1) \wedge (z_1 dz_1) = d\bar{z}_1 dz_1 dw_1 d\bar{w}_1 dt_1 \left(|z_1|^2 |w_1|^2 \bar{z}_2\right) \times
$$
$$
\left[ \partial_{\bar{z}_0}\left( \frac{\bar{z}_{01}\bar{w}_{12}dt_2 - \bar{z}_{01}t_{12}d\bar{w}_2 + \bar{w}_{01}t_{12}d\bar{z}_2 - \bar{w}_{01}\bar{z}_{12}dt_2 + t_{01}\bar{z}_{12}d\bar{w}_2 - t_{01}\bar{w}_{12}d\bar{z}_{12}}{d_{01}^5 d_{12}^9} \right) \right. \tag{180}
$$
$$
\left. - \frac{\partial_{\bar{z}_0}(\bar{z}_{01}\bar{w}_{12}dt_2 - \bar{z}_{01}t_{12}d\bar{w}_2 + \bar{w}_{01}t_{12}d\bar{z}_2 - \bar{w}_{01}\bar{z}_{12}dt_2 + t_{01}\bar{z}_{12}d\bar{w}_2 - t_{01}\bar{w}_{12}d\bar{z}_{12})}{d_{01}^5 d_{12}^9} \right]
$$

By integration by parts, the the integral over $t_1$, $z_1$, $\bar{z}_1$, $w_1$, $\bar{w}_1$ of all the terms in the first two lines vanishes.

So we are left with

$$
- \int_{v_1} dt_1 dz_1 d\bar{z}_1 dw_1 d\bar{w}_1 \frac{|z_1|^2 |w_1|^2 \bar{z}_2(\bar{w}_{12}dt_2 - t_{12}d\bar{w}_2)}{d_{01}^5 d_{12}^9} \tag{181}
$$

**Lemma 2.**

We can use Feynman integral technique to convert (181) to the following:

$$
\int_{v_1}\int_0^1 dx \frac{\Gamma(7)}{\Gamma(5/2)\Gamma(9/2)} \frac{\sqrt{x^3(1-x)^7}|z_1|^2|w_1|^2\bar{z}_2(\bar{w}_{12}dt_2 - t_{12}d\bar{w}_2)}{((1-x)(|z_1|^2+|w_1|^2+t_1^2) + x(|z_{12}|^2+|w_{12}|^2+t_{12}^2))^7}
$$
$$
= \int_{v_1}\int_0^1 dx \frac{(\Gamma \text{ factors})\sqrt{x^3(1-x)^7}|z_1|^2|w_1|^2\bar{z}_2(\bar{w}_{12}dt_2 - t_{12}d\bar{w}_2)}{(|z_1 - xz_2|^2+|w_1 - xw_2|^2 + (t_1 - xt_2)^2 + x(1-x)(|z_2|^2+|w_2|^2 + t_2^2))^7} \tag{182}
$$

Shift the integral variables as

$$
z_1 \to z_1 + xz_2, \quad w_1 \to w_1 + xw_2, \quad t_1 \to t_1 + xt_2 \tag{183}
$$

Then the above becomes

$$
\int_{v_1}\int_0^1 dx \frac{\Gamma(7)}{\Gamma(5/2)\Gamma(9/2)} \frac{\sqrt{x^3(1-x)^7}|z_1 + xz_2|^2|w_1 + xw_2|^2\bar{z}_2}{(|z_1|^2+|w_1|^2+t_1^2 + x(1-x)(|z_2|^2+|w_2|^2 + t_2^2))^7}
$$
$$
\times ((\bar{w}_1 + (x-1)\bar{w}_2)dt_2 - (t_1 + (x-1)t_2)d\bar{w}_2) \tag{184}
$$

Drop terms with odd number of $t_1$ and terms that has holomorphic or anti-holomorphic dependence on $z_1$ or $w_1$:

$$
\int_{v_1}\int_0^1 dx \frac{\Gamma(7)}{\Gamma(5/2)\Gamma(9/2)} \frac{\sqrt{x^3(1-x)^9}(|z_1|^2 + x^2|z_2|^2)(|w_1|^2 + x^2|w_2|^2)\bar{z}_2(\bar{w}_2 dt_2 - t_2 d\bar{w}_2)}{(|z_1|^2+|w_1|^2 + t_1^2 + x(1-x)(|z_2|^2+|w_2|^2 + t_2^2))^7} \tag{185}
$$

After doing the $v_1$ integral using Mathematica with the integral measure $dt_1 dz_1 d\bar{z}_1 dz_2 d\bar{z}_2$, we get

$$
\bar{z}_2(\bar{w}_2 dt_2 - t_2 d\bar{w}_2)\left( \frac{c_1}{d_{02}^5} + \frac{c_2 w_2^2}{d_{02}^7} + \frac{c_3 z_2^2}{d_{02}^7} + \frac{c_4 z_2^2 w_2^2}{d_{02}^9} \right) \tag{186}
$$

**Lemma 3.**

We will compute the integral over the second vertex.

$$
\int_{v_2} \mathcal{P} \wedge \partial_{w_2} P_3(v_2, v_3) \wedge dz_2 \wedge dw_2(z_2 w_2^2 \partial_{z_2}\partial_{w_2}^2 A)
$$
$$
= \int_{v_2} \mathcal{P} \wedge \frac{\bar{w}_2(\bar{z}_{23}d\bar{w}_2 dt_2 - \bar{w}_{23}d\bar{z}_2 dt_2 + t_{23}d\bar{z}_2 d\bar{w}_2)}{d_{23}^7} \wedge dw_2 \wedge dz_2 \tag{187}
$$

Now, compute the integrand:

$$
\frac{\bar{z}_2(\bar{w}_2 dt_2 - t_2 d\bar{w}_2)\bar{w}_2(\bar{z}_{23}d\bar{w}_2 dt_2 - \bar{w}_{23}d\bar{z}_2 dt_2 + t_{23}d\bar{z}_2 d\bar{w}_2)}{d_{02}^5 d_{23}^7} \wedge dw_2 \wedge dz_2
$$

$$
= \frac{|z_2|^2 |w_2|^4 (t_2 - t_3 - t_2)}{d_{02}^5 d_{23}^7} dt_2 d\bar{z}_2 d\bar{w}_2 dw_2 dz_2
$$

$$
= -\frac{|z_2|^2 |w_2|^4 t_3}{d_{02}^5 d_{23}^7} dt_2 d\bar{z}_2 d\bar{w}_2 dw_2 dz_2 \quad \text{substitute } t_3 = \epsilon, \text{ then,}
$$

$$
= -\frac{|z_2|^2 |w_2|^4 \epsilon}{d_{02}^5 d_{23}^7} dt_2 d\bar{z}_2 d\bar{w}_2 dw_2 dz_2
$$

(188)

We can rescale $\epsilon \to 1$, without loss of generality, then it becomes

$$
-\frac{|z_2|^2 |w_2|^4}{d_{02}^5 d_{23}^7} dt_2 d\bar{z}_2 d\bar{w}_2 dw_2 dz_2
$$

(189)

**Lemma 4.**
Now, it remains to evaluate the delta function at the third vertex. In other words, substitute:

$$
w_3 \to 0, \quad z_3 \to 0, \quad t_3 \to \epsilon = 1
$$

(190)

Then, use Feynman technique to convert the above integral into

$$
-\frac{\Gamma(6)}{\Gamma(5/2)\Gamma(7/2)} \int_0^1 dx \int_{v_2} \frac{\sqrt{x^3(1-x)^5}|z_2|^2|w_2|^4}{(x(z_2^2 + w_2^2 + (t_2-1)^2) + (1-x)(z_2^2 + w_2^2 + t_2^2))^6}
$$

$$
= -\frac{\Gamma(6)}{\Gamma(5/2)\Gamma(7/2)} \int_0^1 dx \int_{v_2} \frac{\sqrt{x^3(1-x)^5}|z_2|^2|w_2|^4}{(z_2^2 + w_2^2 + (t_2-x)^2 + x(1-x))^6}
$$

$$
= -\frac{\Gamma(6)}{\Gamma(5/2)\Gamma(7/2)} \int_0^1 dx \int_{v_2} \frac{\sqrt{x^3(1-x)^5}|z_2|^2|w_2|^4}{(z_2^2 + w_2^2 + t_2^2 + x(1-x))^6}
$$

(191)

In the second equality, we shifted $t_2$ to $t_2 + x$.

After doing $v_2$ integral, it reduces into

$$
\frac{\Gamma(6)}{\Gamma(5/2)\Gamma(7/2)} \frac{\pi}{2880} \int_0^1 dx\, x(1-x)^2 = \frac{\Gamma(6)}{\Gamma(5/2)\Gamma(7/2)} \frac{\pi}{2880}
$$

(192)

Finally, re-introduce all the omitted constants:

$$
(FirstTerm) = \frac{\Gamma(6)}{\Gamma(5/2)\Gamma(7/2)} \frac{\Gamma(7)}{\Gamma(5/2)\Gamma(9/2)} (2\pi)^2 (2\pi)^2 \frac{\pi}{2880}
$$

(193)

Similarly, we can compute all the others without any divergence.

$$
(\text{Second Term}) = \frac{\Gamma(6)}{\Gamma(5/2)\Gamma(7/2)} \frac{\Gamma(7)}{\Gamma(5/2)\Gamma(9/2)} (2\pi)^2 (2\pi)^2 \frac{\pi}{5760}
$$

$$
(\text{Third Term}) = \frac{\Gamma(6)}{\Gamma(5/2)\Gamma(7/2)} \frac{\Gamma(7)}{\Gamma(5/2)\Gamma(9/2)} (2\pi)^2 (2\pi)^2 \frac{\pi}{8640}
$$

(194)

$$
(\text{Fourth Term}) = \frac{\Gamma(6)}{\Gamma(5/2)\Gamma(7/2)} \frac{\Gamma(7)}{\Gamma(5/2)\Gamma(9/2)} (2\pi)^2 (2\pi)^2 \frac{\pi}{20160}
$$

Hence, every terms in (186) are integrated into finite terms.

## B.2   Intermediate steps in section 5.2

**Lemma 5.**

We want to evaluate the following integral.

$$\int_{v_1} \partial_{z_1} P_1(v_0, v_1) \wedge (w_1 dw_1) \wedge (z_1^2 dz_1) \wedge \partial_{w_1} P_2(v_1, v_2) \tag{195}$$

Substituting the expressions for propagators, we get

$$\int_{v_1} \frac{|z_1|^2 z_1 w_1 (\bar{w}_1 - \bar{w}_2)}{d_{01}^7 d_{12}^7} (\bar{z}_{01} \bar{w}_{12} dt_2 - \bar{z}_{01} t_{12} d\bar{w}_2 + \bar{w}_{01} t_{12} d\bar{z}_2 - \bar{w}_{01} \bar{z}_{12} dt_2$$
$$+ t_{01} \bar{z}_{12} d\bar{w}_2 - t_{01} \bar{w}_{12} d\bar{z}_2) d\bar{z}_1 d\bar{w}_1 dt_1 dz_1 dw_1 \tag{196}$$

We already know that the terms proportional to $\bar{w}_2$ will vanish in the second vertex integral, so drop them. Evaluating the delta function at $v_0$, the above simplifies to

$$\int_{v_1} \frac{|z_1|^2 z_1 |w_1|^2}{d_{01}^7 d_{12}^7} \left( - \bar{z}_1 \bar{w}_{12} dt_2 + \bar{z}_1 t_{12} d\bar{w}_2 - \bar{w}_1 t_{12} d\bar{z}_2 + \bar{w}_1 \bar{z}_{12} dt_2 \right.$$
$$- t_1 \bar{z}_{12} d\bar{w}_2 + t_1 \bar{w}_{12} d\bar{z}_2) d\bar{z}_1 d\bar{w}_1 dt_1 dz_1 dw_1 \tag{197}$$

Note that the integrand with the odd number of $t_1$ vanishes, so

$$\int_{v_1} \frac{|z_1|^2 z_1 |w_1|^2}{d_{01}^7 d_{12}^7} (-\bar{z}_1 \bar{w}_{12} dt_2 - \bar{z}_1 t_2 d\bar{w}_2 + \bar{w}_1 t_2 d\bar{z}_2 + \bar{w}_1 \bar{z}_{12} dt_2) d\bar{z}_1 d\bar{w}_1 dt_1 dz_1 dw_1 \tag{198}$$

Now, apply Feynman technique, and omit the Gamma functions, to be recovered at the end.

$$\int_0^1 dx \sqrt{x(1-x)}^7 \int_{v_1} \frac{|z_1|^2 |w_1|^2 z_1 (-\bar{z}_1 \bar{w}_{12} dt_2 - \bar{z}_1 t_2 d\bar{w}_2 + \bar{w}_1 t_2 d\bar{z}_2 + \bar{w}_1 \bar{z}_{12} dt_2)}{(x(|z_1|^2 + |w_1|^2 + |t_1|^2) + (1-x)(|z_{12}|^2 + |w_{12}|^2 + |t_{12}|^2))^7}$$
$$= \int_0^1 dx \sqrt{x(1-x)}^7 \int_{v_1} \frac{|z_1|^2 |w_1|^2 z_1 (-\bar{z}_1 \bar{w}_{12} dt_2 - \bar{z}_1 t_2 d\bar{w}_2 + \bar{w}_1 t_2 d\bar{z}_2 + \bar{w}_1 \bar{z}_{12} dt_2)}{(|z_1 - xz_2|^2 + |w_1 - xw_2|^2 + (t_1 - xt_2)^2 + x(1-x)(|z_2|^2 + |w_2|^2 + t_2^2))^7} \tag{199}$$

Shift the integral variables as

$$z_1 \to z_1 + xz_2, \quad w_1 \to w_1 + xw_2, \quad t_1 \to t_1 + xt_2 \tag{200}$$

Then the above becomes

$$\int_0^1 dx \sqrt{x(1-x)}^7 \int_{v_1} dz_1 d\bar{z}_1 dw_1 d\bar{w}_1 dt_1 (|z_1|^2 + x^2 |z_2|^2)(|w_1|^2 + x^2 |w_2|^2)(z_1 + xz_2)$$
$$\left( \frac{-(\bar{z}_1 + x\bar{z}_2)(\bar{w}_1 + (x-1)\bar{w}_2)dt_2 - (\bar{z}_1 + x\bar{z}_2)t_2 d\bar{w}_2}{(|z_1|^2 + |w_1|^2 + t_1^2 + x(1-x)(|z_2|^2 + |w_2|^2 + t_2^2))^7} \right.$$
$$\left. + \frac{(\bar{w}_1 + x\bar{w}_2)t_2 d\bar{z}_2 + (\bar{w}_1 + x\bar{w}_2)(\bar{z}_1 + (x-1)\bar{z}_2)dt_2}{(|z_1|^2 + |w_1|^2 + t_1^2 + x(1-x)(|z_2|^2 + |w_2|^2 + t_2^2))^7} \right) \tag{201}$$

The terms with (anti)holomorphic dependence on complex coordinates drop:

$$\int_0^1 dx \sqrt{x(1-x)}^7 \int_{v_1} dz_1 d\bar{z}_1 dw_1 d\bar{w}_1 dt_1 (|z_1|^2 + x^2 |z_2|^2)(|w_1|^2 + x^2 |w_2|^2)$$
$$\left( \frac{-|z_1|^2 t_2 d\bar{w}_2 + x|z_1|^2 \bar{w}_2 dt_2 - x^2 |z_2|^2 (x-1)\bar{w}_2 dt_2}{(|z_1|^2 + |w_1|^2 + t_1^2 + x(1-x)(|z_2|^2 + |w_2|^2 + t_2^2))^7} \right.$$
$$\left. + \frac{-x^2 |z_2|^2 t_2 d\bar{w}_2 + x^2 z_2 \bar{w}_2 t_2 d\bar{z}_2 + x^2 |z_2|^2 \bar{w}_2 (x-1) dt_2}{(|z_1|^2 + |w_1|^2 + t_1^2 + x(1-x)(|z_2|^2 + |w_2|^2 + t_2^2))^7} \right) \tag{202}$$

We can be prescient again; using the fact that the second vertex is tagged with a delta function $\delta(z_2 = 0, t_2 = \epsilon) \propto dz_2 d\bar{z}_2 dt_2$, we can drop most of the terms.

$$
-\int_0^1 dx \sqrt{x(1-x)}^7 \int_{v_1} [dV_1] \frac{(|z_1|^2 + x^2|z_2|^2)(|w_1|^2 + x^2|w_2|^2)(-|z_1|^2 - x^2|z_2|^2)t_2 d\bar{w}_2}{(|z_1|^2 + |w_1|^2 + t_1^2 + x(1-x)(|z_2|^2 + |w_2|^2 + t_2^2))^7}
$$
$$
= -\int_0^1 dx \sqrt{x(1-x)}^7 \int_{v_1} [dV_1] \frac{(|z_1|^2 + x^2|z_2|^2)^2(|w_1|^2 + x^2|w_2|^2)t_2 d\bar{w}_2}{(|z_1|^2 + |w_1|^2 + t_1^2 + x(1-x)(|z_2|^2 + |w_2|^2 + t_2^2))^7}
$$
$$(203)$$

where $[dV_1]$ is an integral measure for $v_1$ integral.

### B.3 Intermediate steps in section 5.3

**Lemma 6.**
We will evaluate the following integral.

$$
\int_{v_1} \frac{1}{w_1} (w_1 dw_1) \delta(t_1 = 0, z_1 = 0) \wedge \partial_{z_2} P_{12}(v_1, v_2)
$$
$$(204)$$

Substituting the expressions for propagators, we get

$$
\int_{v_1} \frac{\bar{z}_1 - \bar{z}_2}{d_{12}^7} (\bar{z}_{12} d\bar{w}_{12} dt_{12} - \bar{w}_{12} d\bar{z}_{12} dt_{12} + t_{12} d\bar{z}_{12} d\bar{w}_{12}) dw_1 \delta(t_1 = z_1 = 0)
$$
$$
= \int_{v_1} \frac{\bar{z}_1 - \bar{z}_2}{d_{12}^7} (\bar{z}_2 d\bar{w}_1 dt_2 + t_2 d\bar{z}_2 d\bar{w}_1) dw_1 \delta(t_1 = z_1 = 0)
$$
$$
= (t_2 d\bar{z}_2 + \bar{z}_2 dt_2) \int_{v_1} \frac{\bar{z}_1 - \bar{z}_2}{\sqrt{t_{12}^2 + |z_{12}|^2 + |w_{12}|^2}^7} d\bar{w}_1 dw_1 \delta(t_1 = z_1 = 0)
$$
$$
= (t_2 d\bar{z}_2 + \bar{z}_2 dt_2) \int dw_1 d\bar{w}_1 \frac{-\bar{z}_2}{\sqrt{t_2^2 + |z_2|^2 + |w_1 - w_2|^2}^7}
$$
$$
= -(t_2 d\bar{z}_2 + \bar{z}_2 dt_2) \int r dr d\theta \frac{\bar{z}_2}{\sqrt{t_2^2 + |z_2|^2 + r^2}^7} = -\frac{2\pi(t_2 d\bar{z}_2 + \bar{z}_2 dt_2)\bar{z}_2}{5\sqrt{t_2^2 + |z_2|^2}^5}
$$
$$(205)$$

where the first equality comes from the fact that $\delta(t_1 = z_1 = 0) \propto dt_1 dz_1 d\bar{z}_1$.

**Lemma 7.**
We will evaluate the following integral.

$$
\int_{v_3} \frac{1}{w_3} (dw_3) \delta(t_3 = 0, z_3 = 0) \wedge \partial_{w_2} P(v_2, v_3)
$$
$$(206)$$

Substituting the expressions for propagators, we get

$$
\int_{v_3} \frac{\bar{w}_2 - \bar{w}_3}{w_3 d_{23}^7} (\bar{z}_{23} d\bar{w}_{23} dt_{23} - \bar{w}_{23} d\bar{z}_{23} dt_{23} + t_{23} d\bar{z}_{23} d\bar{w}_{23}) dw_3 \delta(t_3 = z_3 = 0)
$$

$$
= \int_{v_3} \frac{\bar{w}_2 - \bar{w}_3}{w_3 d_{23}^7} (-\bar{z}_2 d\bar{w}_3 dt_2 + t_2 d\bar{z}_2 d\bar{w}_3) dw_3 \delta(t_3 = z_3 = 0)
$$

$$
= (t_2 d\bar{z}_2 - \bar{z}_2 dt_2) \int_{v_3} \frac{\bar{w}_2 - \bar{w}_3}{w_3 \sqrt{t_{23}^2 + |z_{23}|^2 + |w_{23}|^2}^7} d\bar{w}_3 dw_3 \delta(t_3 = z_3 = 0)
$$

$$
= (t_2 d\bar{z}_2 - \bar{z}_2 dt_2) \int dw_3 d\bar{w}_3 \frac{(\bar{w}_2 - \bar{w}_3)/w_3}{\sqrt{t_2^2 + |z_2|^2 + |w_2 - w_3|^2}^7}
$$

$$
= (t_2 d\bar{z}_2 - \bar{z}_2 dt_2) \int dw_3 d\bar{w}_3 \frac{-\bar{w}_3/(w_3 + w_2)}{\sqrt{t_2^2 + |z_2|^2 + |w_3|^2}^7}
$$

$$
= (t_2 d\bar{z}_2 - \bar{z}_2 dt_2) \int_{|w_3| \le |w_2|} dw_3 d\bar{w}_3 \frac{-\bar{w}_3 \left(1 - \frac{w_3}{w_2} + \frac{1}{2!} \frac{w_3^2}{w_2^2} - \dots\right)}{w_2 \sqrt{t_2^2 + |z_2|^2 + |w_3|^2}^7} \tag{207}
$$

$$
+ (t_2 d\bar{z}_2 - \bar{z}_2 dt_2) \int_{|w_3| \ge |w_2|} dw_3 d\bar{w}_3 \frac{-\bar{w}_3 \left(1 - \frac{w_2}{w_3} + \frac{1}{2!} \frac{w_2^2}{w_3^2} - \dots\right)}{w_3 \sqrt{t_2^2 + |z_2|^2 + |w_3|^2}^7}
$$

$$
= (t_2 d\bar{z}_2 - \bar{z}_2 dt_2) \int_{|w_3| \le |w_2|} dw_3 d\bar{w}_3 \left(0 + \frac{-|w_3|^2}{w_2^2 \sqrt{t_2^2 + |z_2|^2 + |w_3|^2}^7} + 0 + 0 + \dots\right)
$$

$$
= (t_2 d\bar{z}_2 - \bar{z}_2 dt_2) \int_0^{|w_2|} r dr d\theta \frac{-r^2}{w_2^2 \sqrt{t_2^2 + |z_2|^2 + r^2}^7}
$$

$$
= -(t_2 d\bar{z}_2 - \bar{z}_2 dt_2) \frac{2\pi}{15 w_2^2} \left(\frac{2}{\sqrt{t_2^2 + |z_2|^2}^3} - \frac{5|w_2|^2 + 2t_2^2 + 2|z_2|^2}{\sqrt{t_2^2 + |z_2|^2 + |w_2|^2}^5}\right)
$$

**Lemma 8.**

We will evaluate

$$
\int_{v_2} dw_2 \wedge dz_2 \wedge d\bar{z}_2 \wedge dt_2 \frac{4\pi^2 t_2 |z_2|^2}{75 w_2 \sqrt{t_2^2 + |z_2|^2}^5} \left(\frac{2}{\sqrt{t_2^2 + |z_2|^2}^3} - \frac{5|w_2|^2 + 2t_2^2 + 2|z_2|^2}{\sqrt{t_2^2 + |z_2|^2 + |w_2|^2}^5}\right). \tag{208}
$$

Assuming the $w_2$ integral domain is a contour surrounding the origin of $w_2$ plane or a path that can be deformed into the contour, we may use the residue theorem for the first term of (208). After doing $w_2$ integral we have

$$
\int_\epsilon^\infty dt_2 \int_{\mathbb{C}_{z_2}} d^2 z_2 \frac{4\pi^2 t_2 |z_2|^2}{75 \sqrt{t_2^2 + |z_2|^2}^5} \frac{2}{\sqrt{t_2^2 + |z_2|^2}^3} = \frac{2\pi^3}{225 \epsilon^2} \tag{209}
$$

Combining with the other diagram with the second vertex in the $t \in [-\infty, -\epsilon]$, we get

$$
\frac{2\pi^3}{225\epsilon^2} - \left(-\frac{2\pi^3}{225\epsilon^2}\right) = \frac{4\pi^3}{225\epsilon^2} \tag{210}
$$

Re-scaling $\epsilon \to 1$, this is finite.

For the second term of (208), let us choose the contour to be a constant radius circle so that $r(\theta) = R$. We need to use an unconventional version of the residue theorem, as the integrand is not a holomorphic function, depending on $|w_2|^2$. Let $w_2 = Re^{i\theta}$, then for a given integrand $f(w_2, \bar{w}_2)$, we have

$$
I = \int_0^{2\pi} d(Re^{i\theta}) f(Re^{i\theta}, Re^{-i\theta}) \tag{211}
$$

Then, $w_2$ integral is evaluated as

$$-\int_0^{2\pi} \frac{d(Re^{i\theta})}{Re^{i\theta}} \frac{4\pi^2 t_2 |z_2|^2}{75\sqrt{t_2^2+|z_2|^2}^5} \frac{5R^2 + 2t_2^2 + 2|z_2|^2}{\sqrt{t_2^2+|z_2|^2 + R^2}^5} = -\frac{8\pi^3 i t_2 |z_2|^2}{75\sqrt{t_2^2+|z_2|^2}^5} \frac{5R^2 + 2t_2^2 + 2|z_2|^2}{\sqrt{t_2^2+|z_2|^2 + R^2}^5} \tag{212}$$

Before evaluating $z_2$ integral, it is better to work without $R$. using the following inequality is useful to facilitate an easier integral:

$$0 < \frac{8\pi^3 i t_2 |z_2|^2}{75\sqrt{t_2^2+|z_2|^2}^5} \left( \frac{5R^2 + 2t_2^2 + 2|z_2|^2}{\sqrt{t_2^2+|z_2|^2 + R^2}^5} \right) < \frac{(8\pi^3 i t_2 |z_2|^2)(2t_2^2 + 2|z_2|^2)}{75(t_2^2+|z_2|^2)^5} \tag{213}$$

Here we used $R \in Real^+$. The left bound is obtained by $R \to \infty$, and the right bound is obtained by $R \to 0$. We only care the convergence of the integral. So, let us proceed with the inequalities.

$$-\frac{4\pi}{192} \frac{8\pi^3 i}{75} \frac{1}{\epsilon^3} < -\int_\epsilon^\infty dt_2 \int_{\mathbb{C}_{z_2}} d^2 z_2 \frac{8\pi^3 i t_2 |z_2|^2}{75\sqrt{t_2^2+|z_2|^2}^5} \left( \frac{5R^2 + 2t_2^2 + 2|z_2|^2}{\sqrt{t_2^2+|z_2|^2 + R^2}^5} \right) < 0 \tag{214}$$

After rescaling $\epsilon \to 1$, we have a finite answer. Combining with the other diagram with the second vertex in the $t \in [-\infty, -\epsilon]$, we get the left bound as

$$-\frac{4\pi}{192} \frac{8\pi^3 i}{75} - \left( \frac{4\pi}{192} \frac{8\pi^3 i}{75} \right) = -\frac{\pi^4 i}{225\epsilon^3} \tag{215}$$

After rescaling $\epsilon_1 \to 1$, this is also finite.

Hence, combining with (210), we get the bound

$$\frac{4\pi^3}{225\epsilon^2} - \frac{\pi^4 i}{225\epsilon^3} < (208) < \frac{4\pi^3}{225\epsilon^2} \tag{216}$$

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
