# Peer review of "Feynman diagrams and $\Omega-$deformed M-theory"

_SciPost Physics_

## Round 1 · Referee Report · Anonymous · 2020-10-26

Strengths
-The paper demonstrates several technical results in a good level of detail. The results are each achieved using two complementary approaches, and the computations presented should be useful for researchers in the field looking for worked examples.
-The paper approaches an interesting question in a burgeoning field, twisted holography, where more results and concrete computations provide valuable data, and a deeper understanding of the proposed correspondence.
Weaknesses
-Since the M2-M5 brane system studied in the paper is unfamiliar to most researchers, the paper may be hard to follow for the non-expert reader. Numerous mathematical results are employed with minimal explanation, and some points which warrant more careful physical explanation are not remarked upon.
-The paper contains many grammatical errors on every page and needs to be proofread carefully before a resubmission.
Report
This paper contains concrete, correct results on a system of physical interest (an example of twisted holography). The techniques used to obtain algebra commutators are sufficiently unfamiliar to the high energy physics community that it should be valuable to have them published, and the computations are presented in a sufficient level of detail to carefully follow along. The paper suffers in its presentation, which somewhat obfuscates the nice results, and should be revised for clarity before publication.
Requested changes
There are numerous minor grammatical errors on every page (too numerous to provide an exhaustive list), and I would recommend that the authors carefully proofread or use an editing program for the entire draft. Here are a number of other points that may help improving the presentation of the work for publication:
1.) The notation used in equation 1 should be explained. It should also be explained more clearly: how the deformation theory argument establishes the algebra isomorphism, what a (partial) physical interpretation of this argument is in terms of coupling to branes/anti-branes, and if any of these ideas are used in the remainder of the paper.
2.) In the introduction, the authors summarize their main results in two pairs of two bullet points (page 3). The authors should make more clear that each pair of bullet points represent two complementary ways of obtaining the same commutation relation (first for the algebra, then for the bimodule). They should also briefly explain how they compute the commutator in the first (algebraic) way, in the first bullet point, to be contrasted with the physical Feynman diagram approach of the second bullet point.
3.) In the introduction and the beginning of section 2, the authors are somewhat imprecise about how they describe Koszul duality and its relation to an isomorphism of operator algebras. Koszul duality relates different algebras, and it is the Koszul dual of the bulk algebra that has generators in one-to-one correspondence with boundary algebra. The authors should clarify exactly what they mean when they refer to Koszul duality providing an algebra isomorphism to avoid confusion. They should clarify to the readers early on which algebras are isomorphic and which are Koszul dual.
4.) In footnote 4, the authors presumably meant "NS-R formalism" instead of "Ramond-Ramond formalism", but that the superghost was in the Ramond-Ramond sector. These typos should be corrected.
5.) On the bottom of page 7 there appears to be an extra '-' in L_{V-}O.
6.) In the introduction , the authors describe the Omega-deformed space as C x C rather than TN. Since these directions are topologically twisted, the fact that C x C and TN have different metrics is not important for their purposes. In that case, perhaps they should have TN in the introduction right away.
7.) In equation (10) the authors should note where the z1, z2 coordinates are defined (the coordinates of C^2_NC)
8.) The bottom of page 8 references a result of [4] whereby the closed string modes decouple from the D6 brane modes, so that the D6 brane is the only relevant 'gravitational' sector remaining from the M-theory setup. This is a fairly surprising result from the IIA point of view, and it is important to the logic of the paper, and the authors should elaborate on it.
9.) In equation (24) are there supressed higher-order corrections? If so, add a '+..." (At least there are in the usual Moyal product... if there is a truncation here it should be mentioned around (24)).
10.) In page 10, one should change 'observables' to 'operators' since the operators at high ghost number are not traditional physical observables (at least, before applying the descent procedure).
11.) The beginning of section 2.6 should be clarified somewhat. In particular, large-N should not be necessary for Koszul duality, but it is important for holography. That isn't completely clear as written.
12.) The notation in (52) should be explained or perhaps relegated to a footnote.

---

## Round 1 · Referee Report · Anonymous · 2021-1-22

Report
Topological holography is an exact isomorphism between the operator algebras of gravity and field theory.
A beautiful result is the one in ref.[17-18], which map exactly (via the notion of Koszul duality) the operator algebra of M-theory in the omega background and the operator algebra of the worldvolume theory of M2-branes.
One of the key features of the proof is that one extracts non-perturbative information on the topological sector from a perturbative calculation, namely imposing the vanishing of the BRST variation of certain Feynman diagrams.
The aim of this paper is to discuss and properly realize this duality in the special case when M-theory localizes to 5d U(1) Chern Simons, which is technically different from the generic U(K) case (see ref.[19]).
The authors start with a very detailed introduction to the results of topological holography, that is addressed to a more general audience. This is very welcome, because the field has a very high level of mathematical sophistication that sets a high threshold to non-experts.
Then they proceede to the novel part of their work, which is a detailed analysis of a M2/M5 system in the Ω−deformed and topologically twisted M-theory.
The final result is a realization of the operator algebra living on the M2 brane acting on the operator algebra on the M5 brane.
The commutators of this algebra are is reproduced in the field theory dual via an explicit perturbative Feynman-diagram computation that echos the one of reference [17].
The paper is well written and shows great attention to detail. The problem is well motivated and the final results are interesting.
I recommend it for publication.

---

## Editorial Decision

submission_&_refereeing_history